

# A new age model for the Pliocene of the Southern North Sea Basin: evidence for asynchronous shifts of marine and terrestrial climate

Emily Dearing Crampton-Flood[1]*, Lars J. Noorbergen[2], Damian Smits[1], R. Christine Boschman[1], Timme H. Donders[4], Dirk K. Munsterman[5], Johan ten Veen[5], Francien Peterse[1], Lucas Lourens[1], Jaap S. Sinninghe Damsté[1,3]

[1]Department of Earth Sciences, Utrecht University, Utrecht, The Netherlands

[2]Department of Earth Sciences, VU University Amsterdam, Amsterdam, The Netherlands

[3]Department of Marine Microbiology and Biogeochemistry, NIOZ Royal Netherlands Institute for Sea Research and Utrecht University, The Netherlands

[4]Department of Physical Geography, Utrecht University, Utrecht, The Netherlands

[5]TNO-Geological Survey of the Netherlands, Utrecht, The Netherlands

*Correspondence to*: Emily Dearing Crampton-Flood (emily.dearingcrampton-flood@manchester.ac.uk)

*Present address: School of Earth and Environmental Sciences, The University of Manchester, Manchester, The United Kingdom

## Abstract

The mid-Pliocene Warm Period (mPWP, 3254–3025 ka) represents the most recent interval in Earth's history where atmospheric $CO_2$ levels were similar to today. The reconstruction of sea surface temperatures (SSTs) and modeling studies have shown that global temperatures were 2–4 °C warmer than present. However, detailed reconstructions of marginal seas and/or coastal zones that enable linking climate evolution in the marine realm to that on the continents are lacking. This is in part due to the absence of precise age models for coastal zones, as they are generally dynamic systems with varying sediment and fresh water inputs. Here, we present a multi-proxy record of Pliocene climate change in the coastal Southern North Sea Basin (SNSB) based on the sedimentary record from borehole Hank, the Netherlands. The marginal marine setting of the Hank borehole during the late Pliocene



provides an excellent opportunity to correlate marine and terrestrial signals, due to continental sediment input mainly from the proto-Rhine-Meuse river. We improve the existing low-

resolution palynology-based age model for the Hank borehole using oxygen stable isotope measurements ($\delta^{18}$O) of the endobenthic foraminifera species *Cassidulina laevigata*, integrated with biochrono- and seismostratigraphy. Identification of hiatuses and freshwater effects in the record allows us to accurately isolate glacial-interglacial climate signals that can be linked to a reference global benthic $\delta^{18}$O stack. In tandem with the biostratigraphic age

control this results in an age framework for the SNSB for the Late Pliocene (~3200–2800 ka). Our multi-proxy reconstruction for the mPWP shows a strong agreement between lipid biomarker and palynology-based terrestrial temperature proxies, which suggest a stable climate, 1–2 °C warmer than present. In the marine realm, however, biomarker-based SSTs show a large range of variation (10 °C). Nevertheless, the fluctuation is comparable to other

SST records from the North Atlantic and Nordic Seas, suggesting that a common factor, most likely variations in the North Atlantic Current, exerted a strong influence over SSTs in the North Atlantic at this time.

## 1.  Introduction

The Pliocene epoch (5.33–2.6 Ma) is a frequently targeted interval for palaeoenvironmental reconstructions because it is considered an analogue for future climate change. For example, atmospheric $CO_2$ concentrations (380–420 ppmv; Seki et al., 2010; Zhang et al., 2013) and continental configurations during the Pliocene were largely similar to present. Detailed proxy and model comparisons for the so-called mid-Piacenzian Warm Period (mPWP, 3254–3025

ka) have been the focus of the Pliocene Research, Interpretation and Synoptic Mapping (PRISM) group (Dowsett et al., 2010, 2013), and reveal that global temperatures were on average 2–4 °C warmer than present (Haywood et al., 2013). This makes the mPWP an excellent interval to investigate a warmer world associated with the scenarios for our (near) future summarized by the Intergovernmental Panel on Climate Change (IPCC, 2014).

Our understanding of Pliocene climate is largely based on sea surface temperature (SST) reconstructions (e.g. Dowsett et al., 2012), which indicate that global SSTs were 2–6 °C warmer than present. Only a few temperature records exist for the terrestrial realm (Zagwijn, 1992; Salzmann et al., 2013), which also indicate climate was warmer than present, but these temperatures are less well constrained. There are even fewer studies that examine



the phase relations and amplitude of variability in coupled land-sea changes (e.g. Kuhlmann et al., 2006), although this information is of key interest for understanding heat transport and the hydrological cycle during the Pliocene. Sediments on continental shelves receive inputs from both the terrestrial and marine environment, and would thus contain an archive of land-sea climate evolution. The North Sea Basin shelf is a site that potentially hosts a combined record

of SST evolution and climate change in the North Western (NW) European continent during the Pliocene due to input of terrestrial material by large European rivers and the active subsidence that provides sediment accumulation space (Gibbard, 1988). Moreover, significant warming of its waters since the second half of the 20$^{th}$ century (0.6 °C rise on average in the period 1962–2001; Perry et al., 2005) indicates the sensitivity of the area for recording

climate change. The region has been a type area for Pliocene and early Pleistocene terrestrial stages (see overview in Zagwijn, 1992), but most studied sections lack absolute dating and land-sea correlation as they target deltaic deposits (Donders et al., 2007; Kemna and Westerhoff, 2007). However, the shallow marine deposits of the Southern North Sea Basin (SNSB) allow better chronostratigraphy building through integrated paleomagnetic, isotope,

and biostratigraphic approaches (e.g. Kuhlmann et al., 2006; Noorbergen et al., 2015; Donders et al., 2018).

Despite the promising preconditions that should enable Pliocene climate reconstruction using the sedimentary archive of the SNSB, the generation of an independently calibrated age model for coastal zone sediments is often complicated by complex interactions

between sea level, sediment supply, and biotic factors (e.g. Krantz, 1991; Jacobs, 2008; Noorbergen et al., 2015; Donders et al., 2018), which may alter sedimentation rates or cause hiatuses resulting from periods with erosion or non-deposition. Indeed, the Pliocene SNSB was a dynamic system in which multiple westward advances of the Eridanos and Rhine-Meuse Rivers generated clinoform successions (Jansen et al., 2004; Kuhlmann and Wong,

2008; Harding, 2015), and the sedimentary record thus needs to be critically evaluated on its stratigraphic continuity before it can be compared with records from adjacent areas, such as the Nordic Seas and the North Atlantic. Munsterman (2016) reported a Pliocene-age sequence of coastal marine sediments from Hank, located in the South West of the Netherlands. The current age framework for the sequence is based on first (FODs) and last occurrence dates

(LODs) of dinoflagellate cysts (Dearing Crampton-Flood et al., 2018). Due to the lack of an independent age constraint in the SNSB, FODs and LODs were inferred from those in the Nordic Seas and the North Atlantic, introducing an unknown range of age uncertainty to the



biostratigraphic age model (Dearing Crampton-Flood et al., 2018). Furthermore, the resolution of the age model is too low (9 biostratigraphic age tie points for the interval ~ 4.5–2.5 Ma) to identify possible hiatuses or changes in deposition, preventing comparison of the record to other archives from the Northern Hemisphere.

The established method for age model construction involves measuring the stable oxygen isotope content ($\delta^{18}$O) of foraminiferal tests and matching the variability to a global benthic $\delta^{18}$O reference stack, such as LR04 (Lisiecki and Raymo, 2005). However, in more coastal settings this method is complicated due to isotopically lighter fresh water input, which alters the $\delta^{18}$O value of the foraminifera tests (Delaygue et al., 2001; Lubinski et al., 2001). Recently, Noorbergen et al. (2015) were successful in creating a tuned age model for the early Quaternary shallow marine interval of borehole Noordwijk, also located in the SNSB, using the $\delta^{18}$O values of particularly the endobenthic foraminifera (*Bulimina aculeata*, *Cassidulina laevigata*, and *Elphidiella hannai*). The depth habitat of endobenthic foraminifera in the sediment shelters these species from disturbances, such as reworking by bottom currents and freshwater input. Although vital and microhabitat effects still influenced the absolute $\delta^{18}$O values of these foraminifera and caused an offset towards more positive values, the trends in $\delta^{18}$O at Noordwijk clearly resembled those of LR04 (Lisiecki and Raymo, 2005; Noorbergen et al., 2015).

In this study, we follow the approach of Noorbergen et al. (2015) and use $\delta^{18}$O values of the endobenthic foraminifera *Cassidulina laevigata* in the Hank borehole to improve the current low-resolution biostratigraphic age model for the Pliocene SNSB of Dearing Crampton-Flood et al. (2018). Reconstruction of the age model is further supported by the identification of hiatuses based on seismic information and gamma ray logging. Subsequently, we complement the existing terrestrial air temperature record for Pliocene NW Europe based on soil bacterial membrane lipid distributions stored in the Hank sediments (from Dearing Crampton-Flood et al., 2018), with multi-proxy records of SST, relative land cover, and terrestrial input based on lipid biomarker proxies, pollen, and dinoflagellate cysts. This enables us to for the first time directly compare marine and terrestrial climate evolution of the SNSB and continental NW Europe during the mid-Piacenzian Warm Period.

2. **Methods**





**2.1 Geological setting and study site**

The Pliocene North Sea was confined by several landmasses, except towards the North, where
it opened into the Atlantic (Ziegler, 1990). At times, there may have also been a connection
via the English Channel to the North Atlantic (Funnel, 1996). In addition to a main marine
water supply via the North Atlantic, the Eridanos River, draining the Fennoscandian shield,
and the Rhine-Meuse River, draining North Western Europe delivered freshwater to the North

Sea (Fig. 1). The proto-Rhine-Meuse river system existed for a large part of the Pliocene,
although it did not drain the Alps until the latest Pliocene (Boenigk, 2002). During the
Pliocene, the sediment supply by the Eridanos River system to the southern area of the SNSB
was limited, such that the Rhine-Meuse river system was the predominant source of sediments
in the Roer Valley Rift system (Westerhoff, 2009). The water depth of the North Sea during

the Pliocene and the Pleistocene was approximately 100 – 300 m in the central part of the
basin (Donders et al., 2018).

The study site (51°43′N, 4°55′E) is located within the current Rhine-Meuse-Scheldt
delta in the municipality of Hank, the Netherlands. The Hank site is located within the Roer
Valley Rift: a region that experienced relatively high tectonic subsidence during the late

Cenozoic (Van Balen et al., 2000). The current drainage area of the Rhine-Meuse-Scheldt
river system is 221,000 km$^2$, however it was likely smaller in the Pliocene (van den Brink et
al., 1993; Boenigk, 2002). Air-lifting well technology was used to drill the Hank borehole
(B44E0146) to a base of 404 m in 2001. Intervals were drilled every 1 m, such that each
sample taken from the meter intervals is an integrated mixture. One advantage of this drilling

technique is that it leads to smoothed records. The gamma ray log of the borehole is readily
accessible from an online database (dinoloket.nl). In addition, a seismic section is available
and covers an east-west transect of the River Meuse (Maas2002 survey, nlog.nl). The
lithology of the Hank borehole (Fig. 2a) is described by the Geological Survey of the
Netherlands (TNO) and Dearing Crampton-Flood et al. (2018). In short, the base of the

succession corresponds to the upper part of the shallow marine Breda Formation, followed by
the sandy, occasionally silty and clay-rich marine delta front deposits sometimes containing
shell fragments, or so-called 'crags', belonging to the Oosterhout Formation. The overlying
Maassluis Formation contains silty shell bearing deltaic to estuarine deposits. For this study,
the interval 404 – 136 m was considered, covering ~4.5 to ~2.5 Myr based on the

biostratigraphic age model of Dearing Crampton-Flood et al. (2018).



## 2.2 Stable isotopes

Sediment samples (n = 269) from the interval between 404 and 136 m were washed and
passed over a series of sieves, after which the >125 μm and >63 μm fractions were collected
and dried at 40 °C. Well preserved foraminifera of the endobenthic species *Cassidulina
laevigata* (i.e. shiny tests) of around the same size were picked from the >125 μm fraction.
Due to the scarcity of foraminifera in some samples, tests were left uncrushed in order to
conserve enough material for isotope analysis. The foraminifera were washed ultrasonically in
water before weighing, and between 10 and 60 μg of intact tests were weighed per sample.
The $\delta^{18}O$ and $\delta^{13}C$ values were measured on a Thermo Gas Bench II (Thermofisher
Scientific) connected to a Delta V mass spectrometer. An in-house NAXOS standard and an
internationally accepted NBS-19 standard ($\delta^{18}O$=-2.20‰, $\delta^{13}C$=1.95‰) were used to calibrate
measured isotope ratios to the Vienna Pee Dee Belemnite (VPDB) standard. Oxygen isotope
ratios were calculated according to the following equation (replacing $\delta^{18}O$ by $\delta^{13}C$ for the
calculation of carbon isotope ratios):

$$\delta^{18}O = \frac{\delta^{18}O_s - \delta^{18}O_{standard}}{\delta^{18}O_{standard}} * 1000‰ \tag{1}$$

Where:

-   $\delta^{18}O$ resembled the eventual data (in ‰) used for comparison with the benthic oxygen
    isotope stack (Lisiecki and Raymo; 2005).
-   $\delta^{18}O_s$ was the isotope value of the sample measured by the mass spectrometer.
-   $\delta^{18}O_{standard}$ was the isotope value measured on the standard.

Outliers were identified when measurements exceeded the range of the upper and
lower boundaries of the standard deviation added to and subtracted from a 7 point moving
average of the isotope record. Since the isotope analysis is coupled, the corresponding value
of $\delta^{18}O$ or $\delta^{13}C$ was removed if either value was identified as an outlier.

## 2.3 Palynology

Organic-walled dinoflagellates that form a cyst during their life cycle are referred to as
dinocysts, and they are preserved in sediments (Head, 1996). Dinocyst assemblages in marine





sediments are linked to infer environmental parameters such as temperature and productivity in surface waters (Rochon et al., 1999; Zonneveld et al., 2013), and can be used as such to reconstruct past climate changes in downcore sediment records (e.g. Pross and Brinkhuis, 2005; Hennissen et al., 2014; 2017). Terrestrial palynomorphs are derived from vegetation and are delivered to coastal marine sediments through wind or river runoff. The pollen and

spore (or sporomorph) assemblage in a downcore sediment record like the coastal marine Hank site can thus indicate the type of vegetation in the nearby continent, which can then be used to infer precipitation and/or temperature regimes of the source area (e.g. Heusser and Shackleton, 1979; Donders et al., 2009; Kotthoff et al., 2014).

Standard palynological techniques were used to process 82 selected samples. HCl and

HF digestion followed by 15 μm sieving were carried out according to Janssen and Dammers (2008). Both marine dinocysts and spores were counted under a light microscope at 400x magnification until a minimum of 200 specimens was found. Rare species were identified during a final scan of the microscope slide. For dinocysts, the taxonomy of Williams et al. (2017) is used.

Some dinocyst taxa prefer cooler (sub)polar waters, hence we may take the sum of those taxa and use that as an indicator for SST in the SNSB. We calculate % cold-adapted dinocysts as the sum of the following species over the sum of all dinocysts in the Hank borehole: *Bitectatodinium* spp., *Habibacysta tectata, Filisphaera filifera, Headinium* spp*., Filisphaera* spp*., Islandinium* spp*., Habibacysta* spp*., Islandinium euaxum,* and *Bitectatonium*

*tepikiense* following the approach adopted by Versteegh and Zonneveld (1994), Donders et al. (2009; 2018), De Schepper et al. (2011).

A subset of 25 samples was analysed for detailed pollen assemblages to provide independent long-term trends in climate and vegetation cover. Late Neogene pollen types can, in most cases, be related to extant genera and families (e.g. Donders et al., 2009; Larsson et

al., 2011). Percent abundances are calculated based on total pollen and spores excluding bisaccate taxa, freshwater algae, and *Osmunda* spores due to peak abundance in one sample of the latter. Bisaccate pollen abundances are excluded because they are heavily influenced by on- to offshore trends (Mudie and McCarthy, 1994), and therefore do not primarily represent tree abundance.

The terrestrial/marine (T/(T+M)) ratio of palynomorphs takes the sum of all sporomorphs and divides by the sum of all sporomorphs and dinocysts. The sum of





sporomorphs excludes bisaccate taxa. The T/M ratio is commonly used as a relative measure
of sea level variations and therefore distance to the coast (e.g. Donders et al., 2009; Kotthoff
et al., 2014).


## 2.4 Lipid biomarkers and proxies

We use three independent organic temperature proxies for sea surface temperature based on
different lipid biomarkers. The $TEX_{86}$ is a proxy based on the temperature sensitivity of
isoprenoidal glycerol dialkyl glycerol tetraethers (isoGDGTs), membrane lipids of marine
archaea (Schouten et al., 2002). An increase in the relative abundance of isoGDGTs
containing more cyclopentane moieties was found to correlate with SSTs (Schouten et al.,
2002). Several transfer functions exist to translate $TEX_{86}$ values into SSTs (e.g. Kim et al.,
2010; Tierney and Tingley, 2014). Here we use the global core-top calibration of Kim et al.
(2010). Since isoGDGTs are also produced in soils, albeit in minor amounts, they may alter
the marine temperature signal during periods with large contributions from land. The relative
input of (fluvially discharged) terrestrial organic matter (OM) can be determined using the
ratio of branched GDGTs (brGDGTs), which are produced in soils (Sinninghe Damsté et al.,
2000; Weijers et al., 2007) and rivers (Zell et al., 2013), with crenarchaeol, an isoGDGT
exclusively produced by marine Thaumarchaeota (Sinninghe Damsté et al., 2002). This ratio
is quantified in the Branched and Isoprenoid Tetraether (BIT) index (Hopmans et al., 2004),
where high BIT indicates a high continental OM input and a low BIT indicates a
predominantly marine source of OM. A BIT index >0.3 is generally used as a cut-off for the
validity of $TEX_{86}$-based SST estimates (Weijers et al., 2006). Secondly, the $U^{K'}_{37}$ index is
used as a proxy for SST based on the degree of unsaturation of $C_{37}$ alkenones produced by
marine haptophyte algae (Prahl and Wakeham, 1987). An increased abundance of the tri-
relative to the di-unsaturated $C_{37}$ alkenones, expressed as the $U^{K'}_{37}$ index, is linked with
decreasing temperature, an adaptation thought to retain membrane fluidity in cooler
environments (Marlowe et al., 1984). $U^{K'}_{37}$ values can be converted to SSTs using the global
core top calibration of Müller et al. (1998), with a calibration error of 1.5 °C. Finally, SSTs
can be reconstructed based on the relative distribution of long chain diols, which are
dihydroxylated lipids with 22–38 carbon atoms. The $C_{28}$ 1,13- $C_{30}$ 1,13- and $C_{30}$ 1,15 diols are
most commonly found in seawater, and have a putative phytoplankton source (Volkman et al.,
1992; Rampen et al., 2007; Rampen et al., 2011). The distribution of these three diols are used




to formulate the long chain diol index (LDI), which can be converted to SST using the
calibration of Rampen et al. (2012), of which the calibration error is 2.0 °C. Furthermore,
since freshwater eustigmatophyte algae produce $C_{32}$ diols (Volkman et al., 1992, 1999; Gelin
et al., 1997), the percentage of the $C_{32}$ diol versus that of the marine $C_{28}$ 1,13- $C_{30}$ 1,13- and
$C_{30}$ 1,15 diols used in the LDI can be used as an indicator for freshwater discharge (Lattaud et
al., 2017).

Sediments (n = 155) were previously extracted and processed according to procedures
outlined in Dearing Crampton-Flood et al. (2018). The polar fractions, containing GDGTs,
were analysed on an Agilent 1260 Infinity ultra-high performance liquid chromatography
instrument (UHPLC) coupled to an Agilent 6130 single quadrapole mass detector with
settings following Hopmans et al. (2016). Injection volume of each sample was 10 µL.
GDGTs were separated using two silica Waters Acquity UPLC HEB Hilic (1.7 µm 2.1 mm x
150 mm) columns (30 °C). A flow rate of 0.2 ml/min was used for isocratic elution: starting
with 82% A and 18% B for 25 min, then a linear gradient to 70% A and 30% B for 25 min
(A= hexane, B=hexane:isopropanol 9:1, *v/v*). Prior to mass detection, atmospheric pressure
chemical ionisation (APCI) with the following source conditions was used: vaporizer
temperature 400 °C, gas temperature 200 °C, capillary voltage 3500 V, drying gas ($N_2$) flow
6L/min, nebulizer pressure 25 psi, corona current 5.0 µA. Selected ion monitoring (SIM) was
used to detect $[M-H]^+$ ions of the isoGDGTS: *m/z* 1302, 1300, 1298, 1296, 1292.

After GDGT analysis, polar fractions were silylated by the addition of N,O-
bis(trimethylsilyl)trifluoroacetamide (BSTFA) and pyridine (60 °C, 20 min). A Thermo trace
gas chromatograph (GC) coupled to a Thermo DSQ mass spectrometer (MS) was used to
analyse long chain diol distributions in SIM mode (*m/z* 299, 313, 327, 341) at the Royal
NIOZ. The temperature program was: 70 °C for one min, then ramp to 130 °C at 20 °C/min,
then ramp to 320 °C at 4 °C/min, then held for 25 minutes.

Ketone fractions, containing the $C_{37}$ alkenones, were analysed using gas
chromatography with flame ionisation detection (GC-FID). Samples were injected (1 µL)
manually on a Hewlett Packard 6890 series GC system equipped with a CP-Sil-5 fused silica
capillary column (25 m x 0.32 mm, film thickness 0.12 µm) and a 0.53 mm pre-column. The
oven temperature program was similar to that used for long chain diol analysis.



## 3. Results

### 3.1 Stable isotopes of *Cassidulina laevigata*

Foraminifera preservation in the intervals 404–386 and 204–136 meters was either very low or non-existent. Furthermore, foraminifera were challenging to pick in the crag material (220–205 m, Fig. 2c). The $\delta^{18}O_{cass.}$ values of foraminifera in the remaining intervals before and after outlier removal ranges from -1.0–3.2 ‰ and 0.3–2.9 ‰, respectively (Fig. 2c). The variability in $\delta^{18}O_{cass.}$ values between peaks and adjacent troughs in the Hank record ranges from ~ 0.9–1.8 ‰. The stable carbon isotope record varies between -2.2–0.6 ‰ (Fig. 2d). The variability in the $\delta^{13}C_{cass.}$ record ranges from ~0.3–2.3 ‰, although the sample at 206 m is probably an outlier but is not removed because the outlier identifier method calculates the 7-point average of the data series (Sect. 2.2), and thus the sample at 206 m is at the upper limit of the range of the isotope record (382–204 m; Fig. 2) for which no moving average can be calculated. Discounting the sample at 206 m, the variability in the $\delta^{13}C_{cass.}$ record is similar than that of the $\delta^{18}O_{cass.}$ record (~1 ‰, Fig. 2d). The interval with the most outliers for both carbon and oxygen isotopes was 311–303 m, where all samples exceeded the upper and lower range of values calculated using the 7-point average (Sect. 2.2).

### 3.2 Palynology

The palynomorphs in the Hank sediments are well preserved. The borehole can be divided into three main intervals according to the (co)dominance of the marine/terrestrial palynomorphs: 1, 2, and 3, which roughly correspond to the early Pliocene (1), mid-Pliocene (2) and late Pliocene/early Pleistocene (3). In the deepest part of the borehole (404–330 m, 1), the marine component of the palynomorph assemblage clearly exceeds the terrestrial, as evidenced by the low T/M values (Fig. 3c). An isolate sporomorph peak and (sub)polar dinocyst peak is visible at 383 m (Figs. 2f, 3c). Interval 2 from 330–187 m shows a fluctuating ratio between the marine and terrestrial elements (Fig. 3c). The cold-adapted dinocysts also show fluctuations indicating alternating warmer and cooler periods (Fig. 2f). One striking feature is the increase in cold-adapted dinocyst abundance and simultaneous *Osmunda* acme at 305 m (Figs. 2f, 3c). The third interval 3 spans the upper part of the borehole (187–136 m), and sporomorphs in particular dominate the spectra from 187 m upward, visible by the consistently high T/M (Fig. 3c). Interval 3 shows an increased



occurrence of coastal marine genera, like *Lingulodinium*. The increased gamma ray values at ~175 m (Fig 2b) are the result of the abundance of (shell) concretions, not clays, and as such, do not indicate a more distal environment but rather a development toward a more proximal environment. At 154–153 m, the marine indicators in the borehole are reduced to just 0.5 %

of the total sum of palynomorphs. However, the highest abundance of cold- adapted dinocysts, mostly composed of taxa like *Habibacysta tectata*, is at 154 m (Fig. 2f). This depth is also marked by the complete disappearance of several (sub)tropical species from the Pliocene, like the genus *Barssidinium* spp. with an LOD at 157 m (Dearing Crampton-Flood et al., 2018). The uppermost (154–136 m) interval indicates an estuarine to deltaic

environment, due to the presence of freshwater and brackish water algae species *Pediastrum* and *Botryococcus*. In contrast, the freshwater indicators are (almost) absent in intervals 1 and 2. The assemblages of interval 3 are also characterized by a fluctuating abundance of cold-adapted dinocysts (Fig. 2f).

The pollen assemblages are dominated by tree pollen, particularly conifers (*Pinus,*

*Picea, Abies, Taxodioidae*-type (including *Glyptostrobus* and *Taxodium*), *Sciatopitys,* and *Tsuga*), but with increasing proportions of grasses (Poaceae, Cyperaceae), and heath (Ericales) in interval 2, and significantly increased amounts of fern spores from 260 m and up (Fig. S1). The angiosperm tree abundances averages about 20% and shows no significant long-term change towards the top of the sequence (Fig. 3). The angiosperm tree pollen record

is diverse, although few taxa are continuously present, and consists mostly of *Quercus robur*-type with significant proportions of *Pterocarya*, *Fagus, Carpinus* and, above 240 m, *Ulmus* (Fig. S1). The Taxodioidae-type shows a distinct long-term decline superimposed by three shorter minima in the end of interval 2 and beginning of 3, occurring at 205, 235 and 170 m (Fig. 3d).


### 3.3 Lipid biomarkers and proxies

IsoGDGTs are present in high abundances throughout the borehole, as evidenced by the high total organic carbon (TOC)-normalized concentrations of crenarchaeol (0.2–130 µg g$^{-1}$ TOC; Dearing Crampton-Flood et al., 2018; Fig. 3c). SSTs are reconstructed for those sediments

where BIT<0.3, i.e. between 404 and 219 m. The trends in the SSTs calculated from the Bayesian (Tierney and Tingley, 2014) and the Kim et al. (2010) calibrations are the same,



however, the absolute values differ (Fig. S2). $TEX_{86}^{H}$-reconstructed SSTs range between 7 and 13 °C, but do not show a clear trend over time (Fig. 2e).

Alkenones are present in the majority of the samples in the interval 404–250 m, however, they are below the detection limit in many of the overlying sediments from 250–200 m. Alkenones re-emerge in the interval from 197–178 m. The $U^{K'}_{37}$ index values range between 0.30–0.83, and correspond to an SST range of 8–24 °C (Fig. 2e). In the early Pliocene (1), the $U^{K'}_{37}$ SST record shows the largest fluctuations in temperature ($\Delta T=15$ °C), and an average SST of 19 °C. Similar variability ($\Delta T=14$ °C) is observed in interval 2

(middle-late Pliocene), although the average SST drops slightly to 17 °C. Alkenones are present around or below the detection limit in 3 (late Pliocene/early Pleistocene), so no SSTs can be calculated. Notably, $U^{K'}_{37}$ SSTs show a warming of 11 °C during the interval between 290–260 m (Fig. 2e).

Long chain diols used for calculation of the LDI index are below the detection limit in

a large proportion of the Hank borehole. SSTs can be reconstructed for a select few samples in sections 1 and 3, and they show scattered temperatures in a range of 13 °C (Fig. 2e). The sediments in section 2 contain enough diols to enable a semi-continuous SST reconstruction. The range of LDI SSTs in B is 4–18 °C (Fig. 2e). The record shows a strong warming trend of 10–12 °C from 295–263 m, coeval with the trend in the $U^{K'}_{37}$ record (Fig. 2). The $\%C_{32}$ diol is

generally high (~36–57%) in 1 and 2 (Early-Mid Pliocene; Fig 3c), indicating a modest to strong freshwater input (cf. Lattaud et al., 2017). The $\% C_{32}$ diol slightly decreases (10 %) over 294–264 m, indicating a gradually decreasing influence of riverine OM and/or an increase in the abundance of the $C_{28}$ 1,13- $C_{30}$ 1,13- and $C_{30}$ 1,15 marine diols. In section 3, the $\%C_{32}$ diol exhibits a strong increasing (21–59 %) trend (187–136 m; Fig. 3).


## 4. Discussion

### 4.1 Age model reconstruction

#### 4.1.1 Environmental setting and seismic profile

The changes in depositional environment in the Hank borehole from open marine to coastal

marine, and successively estuarine conditions are based upon the BIT, TOC, and $\delta^{13}C_{org}$ records presented in Dearing Crampton-Flood et al. (2018), and the biological changes in the





abundances of typical marine (dinocysts and test linings of foraminifera), estuarine/freshwater algae species, and sporomorph assemblages (Munsterman, 2016), summarized in the T/M ratio. A transition of marine OM during the Pliocene to more terrestrial OM input towards the

Pleistocene starts approximately at 190 m as evidenced by these indicators (Fig. 3c).

The ~15 km east to west seismic profile of the Meuse River, including the location of the Hank borehole, spans a depth of >500 m (Maas2002 survey, nlog.nl; Fig. 4). Comparison of the formations of the Hank borehole with the seismic depth profile in Fig. 4 indicates that the Breda Formation at 404–370 m is characterized by horizontal-reflection patterns, likely

indicating shallow marine conditions. The eastern continuation of the seismic line reveals that these horizontal strata can be interpreted as shelf toesets of westward prograding deltaic clinoforms.

The transition to the overlying Oosterhout Formation is marked by a distinct angular unconformity referred to as the Late Miocene Unconformity (LMU; Munsterman et al.

unpublished data; Fig. 4). The seismic data of the overlying Oosterhout Formation (Mid-Late Pliocene) indicates a twofold subdivision. The lower unit of the subdivision at 370–319 m is characterized by convex downward reflection patterns that correspond to an open marine signature (with corresponding low sedimentation rates). This is confirmed by the transition to finer grained sediments (silts) over 381–352 m attributed to a more distal setting (Fig. 2a).

This interval is also characterized by an increased abundance of dinocysts with a preference for open marine conditions, like the genus *Spiniferites*. The facies of interval 352–338 m indicates shallow to open marine conditions with a temperate to (sub)tropical SST. A coarsening upward trend is corroborated by the gradual decrease in gamma ray values. In the second Oosterhout unit from 319–157 m the environment is shallow marine, and several

stacked clinoform sets are visible in the seismic profile (Fig. 4). The transition between the two Oosterhout units is clearly visible as a downlap surface around 319–313 m that corresponds to a possible hiatus (Fig. 4). Above this transition, an increase in water depth to ~200 m can be deduced from the height of the clinoforms where the topsets represent the fluvial distributary system and the clinoform breaks the coastlines. The topset beds in

borehole Hank show an abundance of shell crag facies material corroborating the near shore setting (Figs. 2a, 4). The stratigraphic stacking is first purely progradational (clinoforms) and changes to both progradational and aggradational higher up, suggesting progressive fluvial influence replacing the marine environment. In the Hank borehole, this change is marked by a distinct clay layer at 292–271 m (Fig. 2a). In the upper part of the second Oosterhout



Formation at depths of 260 m and upwards, the increased proportion of heath and grasses is generally considered indicative of colder and drier terrestrial climate (Faegri et al., 1989; Fig. 3). The decrease in Taxodioidae-type pollen over section B and the Oosterhout to the Maassluis Formations further indicates a cooling terrestrial climate, and is classically recognized as the top Pliocene (Late Reuverien) in the continental zonation of Zagwijn (1960)

although in the onshore type area the sequences are most likely fragmented time intervals bounded by several hiatuses (Donders et al., 2007). Strikingly, both the MAT record (Dearing Crampton-Flood et al., 2018) and the Taxodioidae-type pollen covary throughout the record (Fig. 4d). The pattern follows the same trends as the classic paleotemperatures profiles during the coeval Brunssumian, Reuverian, and Praetiglian terrestrial stages. These

paleotemperatures estimates were largely based on Taxodioidae pollen abundances and other warm-temperate elements (Zagwijn 1960; 1992), but lacked the direct chronological control at the Hank site.

At the transition of the Oosterhout Formation to the Maassluis Formation, concave downward reflection patterns may reflect channel incisions into the topsets of the Oosterhout

Formation (Fig. 4). The Maassluis Formation (late Pliocene-early Pleistocene, <2.6 Ma) is composed of horizontal and channel-like strata in the seismic profile (Fig. 4). The environment of the Maassluis Formation becomes more fluvio-deltaic, characterized by the decreased abundance of dinoflagellate cysts and steep rise in the number of sporomorphs, manifested by the high T/M values (Fig. 3c). Further warm-temperate trees in the Maassluis

interval of the record such as *Carya, Liquidambar, Nyssa* disappeared in NW Europe in the earliest Pleistocene (Donders et al., 2007).

### 4.1.2 Chronological constraints

Based upon the age model of Dearing Crampton-Flood et al. (2018), it is clear that the sample

resolution is too low to resolve a stable isotope tuning on Milankovitch time scales for the older succession including the Breda and lower Oosterhout Formations (404–330 m). In contrast, the sample resolution is sufficient (i.e. < 6 kyr) for the depth interval above 298 m. A dramatic decrease in sedimentation rate in the initial age model (Dearing Crampton-Flood et al., 2018) is coupled with a hiatus indicated by a sequence boundary (SB) in the seismic

profile (Fig. 4) around ~330 m. The biostratigraphic age model places this interval within the scope of the M2 glacial event (~3.3 Ma; Dearing Crampton-Flood, 2018). This is further



supported by the absence of excursions towards heavier values in the $\delta^{18}O_{cass.}$ record for any point in the interval deeper than 300 m (Fig. 2c). Due to the M2 being a globally recognized event (De Schepper et al., 2014), this indicates that a hiatus likely exists over the most acute

part of the glacial. A Pliocene benthic $\delta^{18}O$ record adjacent to the NS in the Nordic Seas (ODP hole 642B; Risebrobakken et al., 2016) also does not record any strong evidence of the M2 event, and the authors postulated that the M2 might have occurred during a hiatus in the borehole. There is evidence for a large sea level drawdown of 70 m in the North Sea (Miller et al., 2011) during the M2 that would have led to a hiatus. In addition, equivocal

temperature/assemblage signals in the Coralline Crag Formation are hypothesized to be a result of sea-level change associated with the M2, which would have decreased or ceased sedimentation entirely (Williams et al., 2009). This indicates that sedimentation in the SNSB was sensitive to disturbance due to its shallow depth. Overall, the combined climate data from the Nordic Seas, East of England, and the Hank site indicate that a significant hiatus (~ 319–

313 m at Hank) occurred in the interconnecting basins. Thus, the coolest interval (with the presumed lowest sea level) of the M2 was not recorded.

The Plio-Pleistocene transition (2.6 Ma) occurs between 200–154 m (Dearing Crampton-Flood et al., 2018). This transition is accompanied by a peak in gamma ray values at ~175 m (Fig. 2b). The upper (Plio-Pleistocene transition) and lower (M2 event) boundaries

identified here provide a contextual framework to construct a higher resolution age model for the mPWP (3254–3025 ka) using stable isotopes of *Cassidulina laevigata*. The open marine signature and relatively horizontally deposited clinoform sets in the second unit of the Oosterhout subdivision from ~305–200 m (Fig. 4) should be suitable for age model reconstruction. However, the coastal marine depositional setting for the Hank borehole in the

chosen interval (upper Oosterhout Formation) during the late Pliocene/Early Pleistocene strongly indicates that successions are likely not continuous, but are stacks representing short time windows (cf. Donders et al., 2007). It is likely that these short time windows share common features, e.g. warmer or cooler intervals. This indicates that tuning the $\delta^{18}O_{cass.}$ record at the Hank borehole to the LR04 stack (Lisiecki and Raymo, 2005) should correlate

either the warmer or cooler intervals.

### 4.1.3 Age model tuning





The absolute values of the oxygen isotope measurements on *Cassidulina laevigata* recorded in Hank are substantially lower by approximately 1–1.5 ‰ than the composite benthic $\delta^{18}O$

values in the LR04 stack (Lisiecki and Raymo, 2005), as well as those of a nearby Pliocene benthic oxygen isotope record from the Nordic Seas (~2–3 ‰; Risebrobakken et al., 2016). The offset in absolute values is unlikely due to a species-dependant effect, as $\delta^{18}O_{cass.}$ values in a nearby Quaternary-age core from Noordwijk (Noorbergen et al., 2015) were comparable to the LR04 stack (Lisiecki and Raymo, 2005). Hence, the relatively low $\delta^{18}O$ values of the

Hank record likely reflect the influence of freshwater input at this site, which is proximal to the mouth of the paleo Rhine (e.g. Delaygue et al., 2001; Lubinski et al., 2001; Fig. 1). Furthermore, the large $\delta^{18}O_{cass.}$ variability in the Hank record (0.9–1.8 ‰) compared to that in the LR04 stack record (0.2–0.7 ‰ during the Pliocene; Lisiecki and Raymo, 2005) indicates that the shallow and relatively fresh(er) North Sea is more sensitive to climate

disturbance than ocean bottom waters. Thus, salinity changes and sensitivity to freshwater input affect the oxygen isotopes incorporated into *Cassidulina* species, regardless of the endobenthic habitat.

For tuning purposes, a detailed understanding of the North Sea hydrogeography and circulation patterns during the Pliocene must be taken into consideration. During cold periods,

the North Sea circulation slows due to the reduced sea level and inflow of Atlantic water (Kuhlmann et al., 2006). Stratification in the North Sea due to freshwater input from rivers combined with the sluggish circulation and weak influence of the Atlantic waters make cooler periods problematic to tune to due to a $\delta^{18}O_{cass.}$ signature that is probably highly localized and erratic. Moreover, Donders et al. (2007) noted that the coldest phase of glacials of the Plio-

Pleistocene climate development of coastal areas in the NS is likely to be marked by substantial hiatuses caused by non-deposition and erosion. During warmer periods, an increased freshwater input from river outflows is also expected, due to the supposedly wetter climate conditions during interglacials. However, Kuhlmann et al. (2006) linked warmer periods in the Pliocene in the central section of the southern North Sea with the occurrence of

*Cassidulina laevigata*, whose habitat in the modern North Sea is located in the northern part with a strong connection to the Atlantic (Murray, 1991). Thus, tuning the warmer periods in the $\delta^{18}O_{cass.}$ record at the Hank site with warm periods in the LR04 benthic stack is preferable due to the strong(er) connection to the Atlantic (Kuhlman, 2004), resulting in a relatively more regional signature of the $\delta^{18}O_{cass.}$ values (Kuhlmann et al., 2006). Moreover, the chance



of disturbance/hiatuses that affect the continuity of the sediment record at Hank is decreased
in warmer periods, thus making them suitable for tuning.

Using the above reasoning, the sample with the lowest $\delta^{18}O_{cass.}$ value in each cycle
between ~300–200 m in the Hank record can be tuned to the lowest $\delta^{18}O$ value between the
M2 and the Plio-Pleistocene boundary (Sect. 4.1.2) in the LR04 stack, presuming that the low
$\delta^{18}O$ values in the Hank borehole represent the warmest part of each interglacial. Further
investigation in the variation of the $\delta^{18}O_{cass.}$ cycles in the Hank borehole isotope record reveals
unique saw tooth structures, representing a different pattern than the more symmetrical pattern
of cyclicity that is seen in the Pleistocene. Specifically, cycles G19, G17, and G15 display
these reversed saw-tooth patterns in the global benthic stack, and help pinpoint corresponding
cycles in the Hank borehole record (Fig. 5). Hence, starting from the initial age constraints,
we correlate lower values in our $\delta^{18}O_{cass.}$ record to those in the global benthic LR04 stacked
record (Fig. 5). The reconstructed time window spans ~3200–2800 ka, and thus most of the
mPWP. Based on the tuned oxygen isotope age model, the LOD of *Invertocysta lacrymosa*
and *Operculodinium? eirikianum* can be constrained to ~3045 and 2600–2782 ka,
respectively (see Dearing Crampton-Flood et al., 2018).

**4.2 SST proxy comparison**

Despite the fact that all three lipid biomarker proxies (TEX$_{86}$, U$^{K'}_{37}$, and LDI) are calibrated
to SST, the records that they generate show remarkable differences and are offset in
temperature (Fig. 2e). Interestingly, the TEX$_{86}^{H}$-derived SST record remains relatively stable
throughout the tuned interval, whereas the U$^{K'}_{37}$ and LDI-based records show large variability
(Fig. 2e). The Pliocene TEX$_{86}^{H}$ SSTs are 10 °C on average, which is the same temperature as
the modern mean SST of the North Sea (Locarnini et al., 2013), and contrasts with other
North Sea Pliocene temperature estimates based on ostracod, mollusc, foraminiferal, and
dinocyst assemblages (Wood et al., 1993; Kuhlman et al., 2006; Johnson et al., 2009;
Williams et al., 2009), all suggesting that the SST of the North Sea was 2–4 °C warmer than
present at that time. Given the shallow water depth of the SNSB in the Pliocene (50–200 m;
Hodgson and Funnel, 1987; Long and Zalasiewicz, 2011) at the Hank site, it seems unlikely
that the isoGDGTs are influenced by the contribution of a subsurface isoGDGT-producing
community. This can be further confirmed by calculating the ratio of isoGDGT-2/isoGDGT-3
([2]/[3]; Taylor et al., 2013), whose value increases with increasing isoGDGT input from



subsurface dwelling archaea. The [2]/[3] ratio in the Hank borehole is 2.1 on average, and always well below the value associated with a deep-water archaea community overprint (>5; Taylor et al., 2013). Instead, the low $TEX_{86}^H$ SSTs are likely a result of seasonal production

of isoGDGTs. In the modern North Sea the main period of Thaumarchaeotal blooms and associated isoGDGT production is in the winter months where ammonia is available and competition with phytoplankton is minimal (Herfort et al., 2006; Pitcher et al., 2011), which likely introduces a cold bias in $TEX_{86}$-based SST estimates for the SNSB.

Conversely, $U^{K'}_{37}$ reconstructed SSTs are 16 °C on average, and thus 2–4 °C higher

than the temperature estimates based on ostracod, mollusc, and foraminiferal assemblages (Wood et al., 1993; Kuhlmann et al., 2006; Johnson et al., 2009; Williams et al., 2009) and ca. 6 °C higher than modern annual mean SST. These higher-than-expected $U^{K'}_{37}$ SSTs could in part be caused by a species effect as a result of a contribution from alkenones produced by freshwater haptophyte algae that have little to no correlation of $U^{K'}_{37}$ with temperature

(Theroux et al., 2010; Toney et al., 2010). Moreover, the influence of freshwater input on salinity may alter the main alkenone producing communities in coastal regions (Fujine et al., 2006; Harada et al., 2008), and thus affect the reliability of SST estimates based on the open ocean calibration specifically adapted for Group III alkenone producers (e.g. *Emiliania huxleyi*). Indeed, strong temperature fluctuations of 10 °C in a Holocene $U^{K'}_{37}$ record from the

Sea of Okhotsk were linked to periods with low sea surface salinity, which were in turn correlated to high $U^{K'}_{37}$-derived SSTs (Harada et al., 2008). In contrast, a recent study showed that alkenone producers in particulate organic matter (POM) in a coastal bay in Rhode Island were unaffected by a lower salinity, further illustrated by the excellent match of the 300-year $U^{K'}_{37}$ SST record with instrumental temperature records, despite the proximity to the river

(Salacup et al., 2019). Although the high variability in the $U^{K'}_{37}$ SST record and the higher-than-expected reconstructed temperatures at Hank fit with a freshwater input as observed in the Sea of Okhotsk, low BIT index values and T/M ratios in the Hank borehole (Fig. 3) suggest that the organic matter has a primarily marine origin. In addition, the absence of the $C_{37:4}$ alkenone in the Hank sediments, a biomarker tentatively linked with coastal or

freshwater haptophytes (Cacho et al., 1999), suggests that the $U^{K'}_{37}$ should mostly represent SSTs. However, a moderate relation between the %$C_{32}$ diol and $U^{K'}_{37}$ derived SST throughout the tuned record (n = 26; $R^2$ = 0.32), suggests that freshwater input may at times have influenced the $U^{K'}_{37}$ SSTs.





Alternatively, the higher $U^{K'}_{37}$ SSTs can be a result of increased production in the

spring or summer (Chapman et al., 1996; Rodrigo-Gámiz et al., 2014). Indeed, summer

temperatures in the Oosterhout Formation (Ouwerkerk, Netherlands) and contemporaneous

Lillo Formation in Belgium (Valentine et al., 2011) recorded from benthic bivalves range

from 14.9–20.4 °C, which is similar to the range of $U^{K'}_{37}$ SSTs in Fig. 2e. This would mean

that summer SSTs were high and very variable during the Pliocene. Although quite variable

in the earlier (~3250–3150 ka) part of the tuned record, $U^{K'}_{37}$ SSTs warmed by approximately

10 °C over the latter part of the tuned interval from 3150 to 3000 ka (Fig. 6d). In line with the

winter-biased and warmer season-biased interpretation of the $TEX_{86}$ and $U^{K'}_{37}$ reconstructed

SSTs, respectively, comparison of the average reconstructed $TEX_{86}$ (10 °C) and $U^{K'}_{37}$ (16°C)

SSTs in the mPWP interval shows good agreement with the PRISM3 model reconstructions

for February (10.4 °C) and August (16.7 °C; Dowsett et al., 2009).

Finally, the LDI record is not complete over the whole depth interval due to low

abundances of long chain diols in various parts of the record. Nevertheless, the LDI-based

SSTs show the same warming trend from ~3150–3000 ka as the $U^{K'}_{37}$ record (Fig. 6d). LDI

SSTs are first 2 °C cooler than the $TEX_{86}$ record, and then increase toward the same SSTs as

in the $U^{K'}_{37}$ record (Fig. 6d). Large discrepancies of 9 °C between $TEX_{86}$ and LDI-derived

SSTs have been observed in the Quaternary of South Eastern Australia (Lopes dos Santos et

al., 2013), which the authors attributed to seasonal production of isoGDGTs in the cooler

months and long chain diols in the warmer months. In late Pliocene sediments from the

central Mediterranean, LDI SST estimates were slightly lower than $U^{K'}_{37}$ SSTs, however this

was within the error range of the proxies (Plancq et al., 2015). Due to the recent advent of the

LDI proxy, and the scarcity of other multi-proxy studies (De Bar et al., 2018; Lattaud et al.,

2018) comparing the LDI to $U^{K'}_{37}$ and $TEX_{86}$ SSTs in the same sediment samples, further

discussion on this topic is limited.

**5. Late Pliocene climate evolution in the southern North Sea Basin**

**5.1 The M2 event and recovery**

The M2 event is most likely incomplete, or absent in the Hank record, and is marked by a

hiatus that is also recognized in sequences from the Coralline Crag in the English North Sea

and the Nordic Seas (Fig. 1, Williams et al., 2009; Risebrobakken et al., 2016). Nevertheless,



in the sediments occurring above the hiatus marked by the sequence boundary in Fig. 4, large

variability in $\delta^{18}O_{cass.}$ indicates fluctuating climate conditions that may be associated with the

onset or the recovery of the M2. The fluctuations match those in the records of the BIT index

and $\delta^{13}C$ of organic matter (see Dearing Crampton-Flood et al., 2018), which indicate a closer

proximity of the coast to the site, likely as a result of sea level change. The major peak of

*Osmunda* spores (outside of pollen percentage sum) after to the hiatus at ~3210 ka (306 m)

could then represent a pioneer phase of marsh vegetation related to a rapid sea level lowering.

The (sub)polar dinocyst acme and increase of *Operculodinium centrocarpum* (Fig. 2, 6a) at

305 m may then represent the restoration of the location of the Hank site to a more distal

marine setting within the confinement of the Rhine Valley Graben. The acme of *Osmunda*

spores coincides with the occurrence of dinocysts characteristic of (sub)polar watermasses at

~3210 ka, further indicating cold conditions (Fig. 6). In addition, the distinct decrease in

Taxodioidae-type pollen at the same time indicates that climate conditions were also cold(er)

on the continent (Fig. 6c), which is supported by low terrestrial mean air temperatures of 6

°C, independently reconstructed based on brGDGTs (Fig. 6f; Dearing Crampton-Flood et al.,

2018). In contrast, all SST reconstructions remain stable during this M2 deglaciation/recovery

period (Fig. 6d), suggesting that cold periods on land are better recorded in the sedimentary

record than those in the marine realm. Indeed, terrestrial proxies represent an integrated signal

over longer time and larger space (NW Europe), compared to that of the marine proxies,

which are confined to the shallow SNSB basin and potentially only record warm periods

(Sect. 4.1.3).

### 5.2 The mid-Piacenzian Warm Period

The mPWP is practically entirely covered by the age-tuned interval of the Hank record, which

starts after the hiatus that marks the M2 event. The sea level drop associated with the M2

event may have decreased the inflow of Atlantic bottom water currents originating from the

Northern opening of the North Sea (Kuhlmann et al., 2006). After the M2 event, isostasy may

have then slowly opened a new connection to the North Atlantic via the English Channel,

which would have allowed the inflow of relatively warmer and saline Atlantic Water fed by

the North Atlantic Current (NAC) into the North Sea (Funnel, 1996). The occurrence of such

an inflow is supported by the high abundance of dinocysts from *Operculodinium*

*centrocarpum*, that is generally used as a tracer for the NAC (Boessenkool et al., 2001; De





Schepper et al., 2009; Fig. 6a), present after the hiatus associated with the M2 event at ~3210 ka (305 m). The amount of *Operculodinium centrocarpum* cysts then decreases to zero and gradually re-emerges from 3150 ka on (Boessenkool et al., 2001; De Schepper et al., 2009; Fig. 6a), indicating a fluctuating influence of the NAC.


The presence of Taxodioidae-type pollen (*Taxodium, Glyptostrobus*) throughout most of the mPWP (Fig. 6c) indicates that land temperatures were generally not low enough for prolonged winter frosts. Minimum Taxodioidae-type pollen abundance of 10% has been associated with a mean temperature of the coldest month of >5 °C (Fauquette et al., 1998).

The terrestrial temperature record of Dearing Crampton-Flood et al. (2018) and the increased proportion of Taxodioidae-type pollen (Fig. 6) support the presumed relatively stable climate conditions on land during the mPWP (Draut et al., 2003; Lisiecki and Raymo, 2005). Importantly, the new chronology for the Hank sediments provides an opportunity to date and quantify the local (Netherlands) qualitative Pliocene-Pleistocene Taxodioidae-type

temperature curves proposed by Zagwijn (1960; 1992). Zagwijn et al. (1992) inferred mean July temperatures between 15–20 °C for the Reuverian, which he placed approximately between 3.1–2.5 Ma, with short lived cool pulses down to ~12 °C that can also be recognized in the brGDGT MAT record (Fig. 6f). Maximum Taxodioidae abundance and mean July temperatures in excess of 20 °C were reconstructed for the Brunssumian placed

approximately between 3.4–3.1 Ma. These reconstructed summer temperatures compare broadly to the SSTs reconstructed using the (presumably) partially summer-biased $U^{K'}_{37}$ proxy, which range between ~10–25 °C in the tuned interval (Fig. 6d). It should be noted that the original terrestrial Pliocene stages as summarized by Zagwijn (1992) have not yet been dated independently, and in the type area of the South East Netherlands, they likely represent

much smaller intervals of time compared to the Hank sequence.

In contrast to the stable terrestrial climate, the LDI and $U^{K'}_{37}$ SST records indicate that SSTs were highly variable during the mPWP (Fig. 6d). The large amplitude of the variation in the SST records may be a result of the at the time relatively shallow coastal location of Hank, which is sensitive to warming and cooling. The proximity to of the site to freshwater input

may also play a role, however this cannot be confirmed (Sect. 4.3). Notably, the high variability in SSTs during the mPWP at the Hank site is also seen in all other currently available $U^{K'}_{37}$ SST records from the North Atlantic (Lawrence et al., 2009; Naafs et al., 2010; Bachem et al., 2017; Clotten et al., 2018). The proposed scenarios for the high





variability in these $U^{K'}_{37}$ SST records range from a change in the strength of the NAC
(Lawrence et al., 2009; Naafs et al., 2010), orbital forcing (Lawrence et al., 2009; Bachem et
al., 2017) and ocean gateway changes (Bachem et al., 2017). In addition, the high variability
in the $U^{K'}_{37}$ record from the Iceland Sea record was linked to the frequent occurrence of
spring sea ice cover and ice-free summers linked to freshwater input (Clotten et al., 2018).
Thus, the high variability of $U^{K'}_{37}$ SSTs at the Hank Site during the Pliocene is most likely
due to a combination of freshwater influence, the shallow depth of the SNSB, and changes in
the direction and strength of the NAC. Orbital forcing may play a role in pacing the variation
of the NAC (Naafs et al., 2010), although this investigation requires further analysis, which is
not possible in the Hank borehole due to the short tuned interval. Nevertheless, the common
factor among the records discussed here is the influence of variations in the position of the
NAC, which thus seems most likely responsible for the variation in all five $U^{K'}_{37}$ SST records
discussed above.

In contrast, the variability of the $U^{K'}_{37}$ SSTs is not reflected in the $TEX_{86}$ record,
which may be due to the winter signal they record (Sect. 4.3). Regardless, a common feature
of the $U^{K'}_{37}$ and LDI SST records is the gradual warming between ~3150–3000 ka (Fig. 6d),
seen most clearly in the LDI record. Before the SST warming from 3200–3100 ka, $\%C_{32}$ diol
decreases slowly (Fig. 6e), indicating a decrease in freshwater discharge and/or an increased
distance to the coast. The low T/M ratios and the presence of a clay layer from 292–271 m in
Fig. 3c (corresponding to 3170–3070 ka; Fig. 6) at this time further indicate increased marine
influence, likely as a result of sea level rise. Differences in the degree of warming recorded by
the organic SST proxies may be attributed to the lateral transport of certain biomarkers
(Benthien and Müller, 2000; Ohkouchi et al., 2002). For example, the change in currents in
the North Sea after the M2 event, bringing in warmer waters from the North Atlantic may
have brought alkenones and/or diols with a warmer signature to the SNSB, resulting in the
high SSTs reflected by the $U^{K'}_{37}$ and LDI proxies. Regardless, the high variability and
warming trend in two out of the three organic SST proxies in the Pliocene North Sea indicate
that the area encompassing the North Atlantic, Nordic Seas, and North Sea was very sensitive
to changing currents, probably as a result of the strength and/or direction of the NAC.





**5. Conclusions**

The age framework for the mid-Pliocene Southern North Sea Basin (SNSB) constructed here reveals that the M2 glacial is represented as a hiatus, confirming interpretations at proximal sites in the Nordic Seas and the English North Sea coast. Our terrestrial multi-proxy records show a highly consistent signal between lipid biomarker temperatures and pollen

assemblages, which show stable terrestrial temperatures of 10–12 °C, and the continued presence of warm-adapted tree species during the mPWP. Importantly, the chronology presented here allows placing earlier terrestrial temperature reconstructions for Pliocene NW Europe (Zagwijn et al., 1992) in time. This indicates that the Reuverian Stage concept, characterized by abundant Taxodioidae and *Sciadopitys* and rare *Sequoia,* is dated to ~3.2–2.8

Ma. Further high-resolution analysis will subdivide and date the Reuverian A–C substages. Conversely, sea surface temperatures were variable, which may be caused by the sensitivity of the shallow Pliocene North Sea to climate change and the influence of freshwater input on lipid biomarker SST proxies. Nevertheless, the variability in SSTs matches that in all other currently available SST records from the North Atlantic and Nordic Seas, indicating that the

marine realm was highly dynamic during the mPWP, probably as a result of shifting currents caused by a reorganization/diversion of the North Atlantic Current after the M2. Thus, our multi-proxy approach for the first time reveals that land-sea climate evolution in the SNSB was asynchronous during the mPWP.

**Data availability**

The research data presented in this paper will soon be available on Pangaea (doi).

**Author contribution**

JSSD, FP, and EDCF designed the research. CB and DS carried out the geochemical analyses

under supervision of EDCF, LN, FP, LL, and JSSD. DM and THD analysed and interpreted the palynological data. JtV provided seismic interpretations. EDCF integrated the data and prepared the manuscript with contributions from all authors.



**Competing interests**

The authors declare that they have no conflict of interest.

**Acknowledgements**

The authors would like to thank Nico Janssen for processing palynological samples, Giovanni Dammers and Natasja Welters for preparation of foraminifera samples, Arnold van Dijk for

help with isotope measurements, and Anita van Leeuwen and Dominika Kasjaniuk for assistance in the organic geochemistry lab. Stefan Schouten and Anchelique Mets at the Royal NIOZ assisted with the analysis of long chain diols. This work was supported by funding from the Netherlands Earth System Science Center (NESSC) through gravitation grant NWO 024.002.001 from the Dutch Ministry for Education, Culture and Science to Jaap S. Sinninghe

Damsté and Lucas Lourens.

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



**Figures**

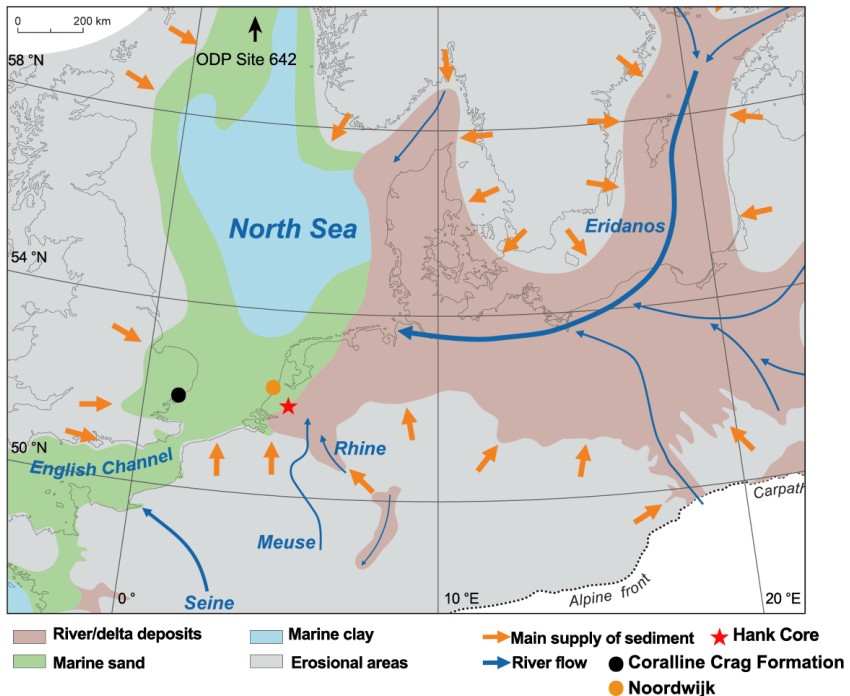

Fig. 1. Pliocene paleogeography in the North Sea basin (Gibbard and Lewin, 2003; Knox et
al., 2010). The location of the Hank borehole is denoted by a red star. Major river and
sediment inputs are represented by blue and orange arrows, respectively. Other locations
mentioned in the text are indicated. Figure modified from Gibbard and Lewin (2016).



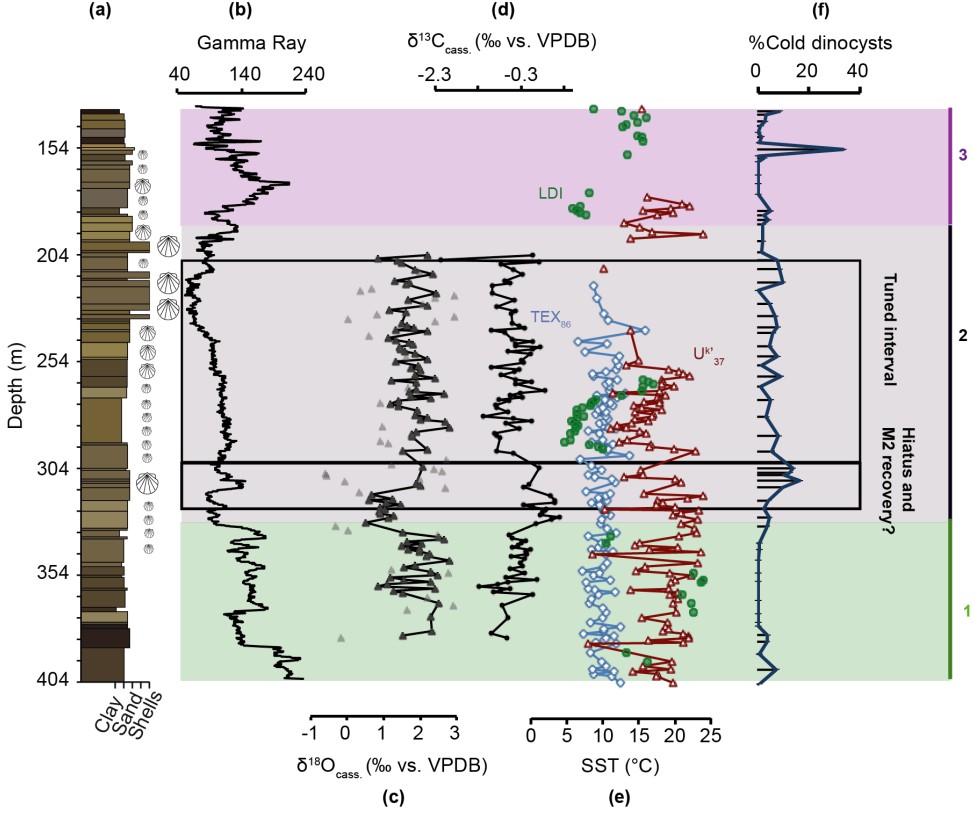

Fig 2. Marine proxies for the Hank borehole. (a) The depth and lithology of the Hank sediments, with shell material qualitatively indicated by shell symbols. (b) The smoothed gamma ray (GR) log (dinoloket.nl). (c) Stable oxygen and (d) stable carbon isotope records for the endobenthic *Cassidulina laevigata*. Outliers in the $\delta^{18}O$ record are indicated by grey triangles and the record after outlier removal is given in black. (e) SST records based on TEX$_{86}$ (blue diamonds), U$^{K'}_{37}$ (red triangles), and LDI (green circles). (f) %Cold taxa of dinoflagellate cysts. The intervals corresponding to the A (green), B (grey), and C (purple) depths discussed in the text are indicated. The tuned interval and the position of the hiatus marking the M2 are represented by a black line.



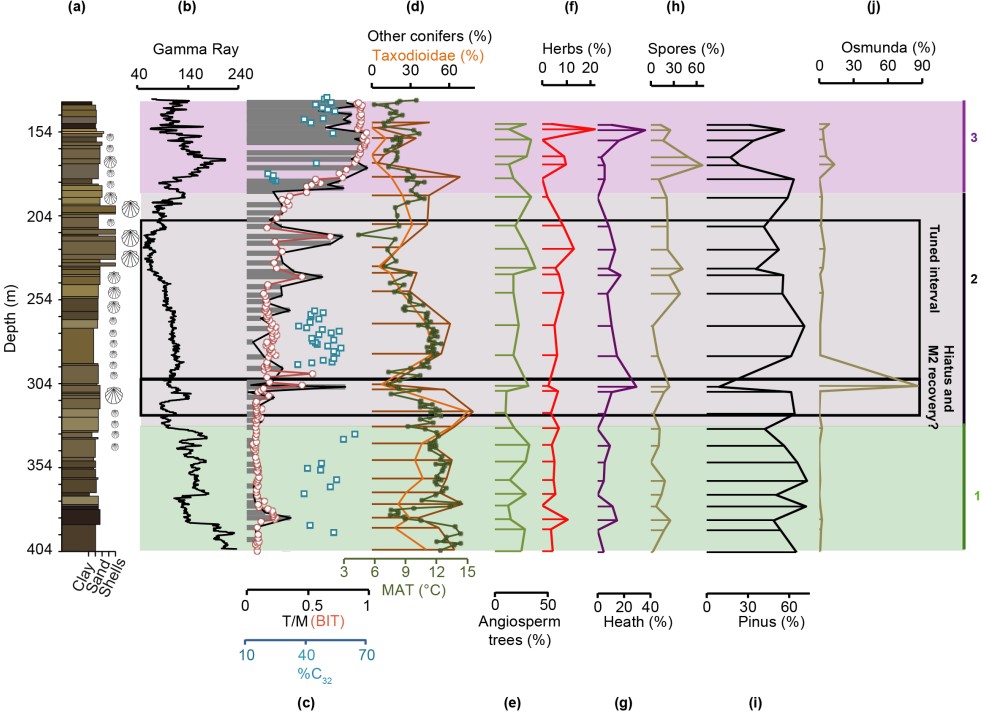


Fig. 3. Terrestrial proxies for the Hank borehole. (a) The depth and lithology of the Hank
sediments, with shell material qualitatively indicated by shell symbols. (b) The smoothed
gamma ray (GR) log (from dinoloket.nl), (c) the relative input of terrestrial organic material
to the Hank sediments based on the terrestrial/marine ratio of palynomorphs (black line), the
Branched and Isoprenoid Tetraether (BIT) index (orange circles), and the %C$_{32}$ diol (blue
squares). Pollen records expressed as % of total pollen: (d) Other conifers (brown line),
Taxodioidae (orange line) and MAT (°C; green line; Dearing Crampton-Flood et al., 2018),
(e) angiosperm trees, (f) herbs, (g) heath, (h) spores, (i) Pinus, and (j) Osmunda. The intervals
corresponding to the A (green), B (grey), and C (purple) depths discussed in the text are
indicated. The tuned interval and the position of the hiatus marking the M2 are represented by
a black line.



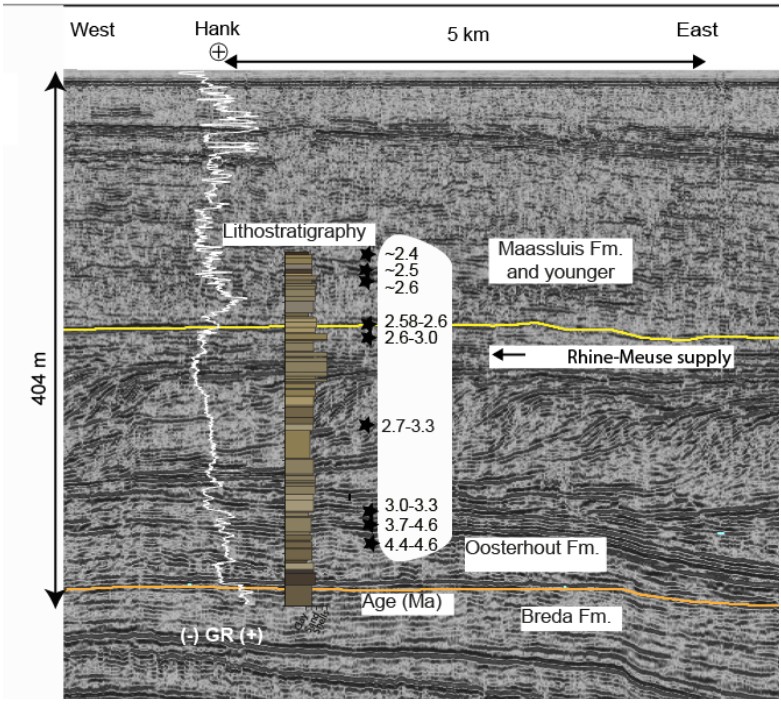

Fig. 4. Seismic east to west depth profile at the River Maas (from Maas2002 survey, nlog.nl)
with the location of the Hank borehole (B44E0146) and corresponding formations indicated.
The smoothed Gamma ray log (from dinoloket.nl, white), and lithology of the borehole are
provided for context. Stars and age ranges refer to the biostratigraphic age model of Dearing
Crampton-Flood et al. (2018). The yellow and orange lines represent the boundaries of the
Breda and Oosterhout (revised), and the Oosterhout and Maassluis formations, respectively.
The blue line represents the depth scale of the borehole (i.e. 404 m).





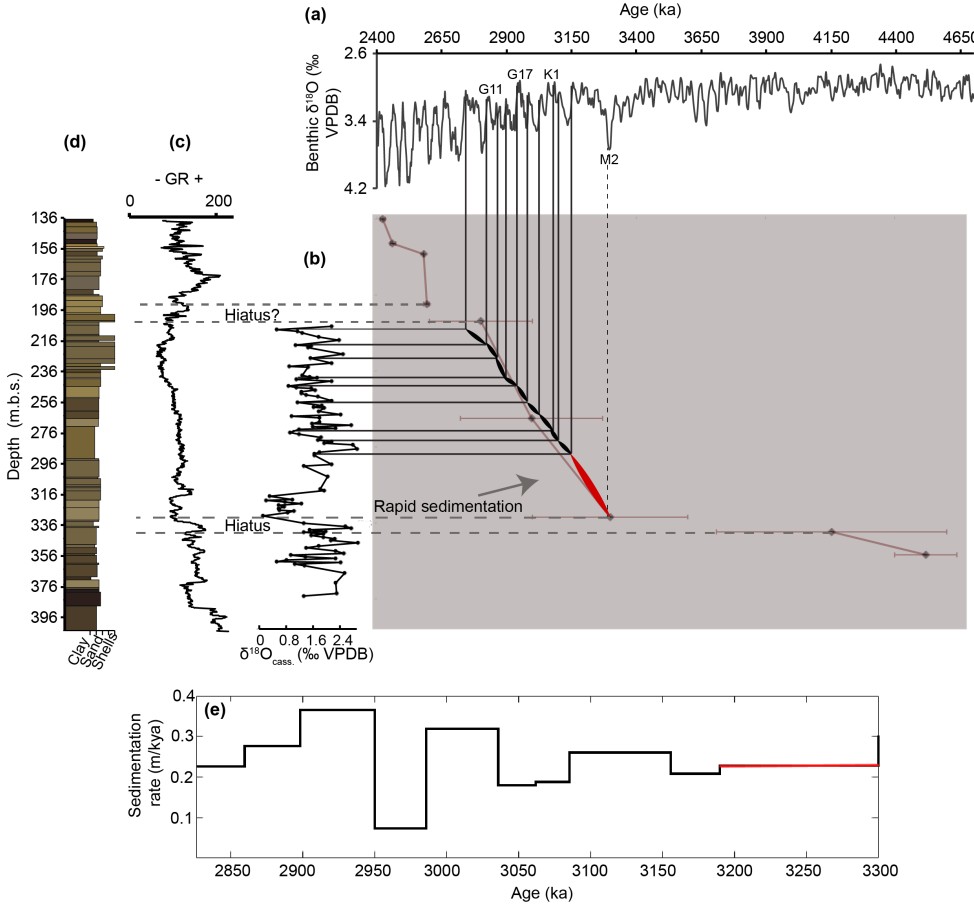

Fig. 5. Age framework for the Pliocene southern North Sea Basin. Correlations of the interglacials of the (a) LR04 stack (Lisiecki and Raymo, 2005) to the interglacials in the (b) $\delta^{18}O_{cass.}$ record for Hank. (c) Smoothed gamma ray (GR), (d) lithology, and depth of Hank sediments. Grey lines indicate tie points based on $\delta^{18}O$ values, whereas the red tie point is based on biostratigraphy. The other biostratigraphic age points (Dearing Crampton-Flood et al., 2018) are given in purple. (e) The sedimentation rate for the tuned interval of Hank.





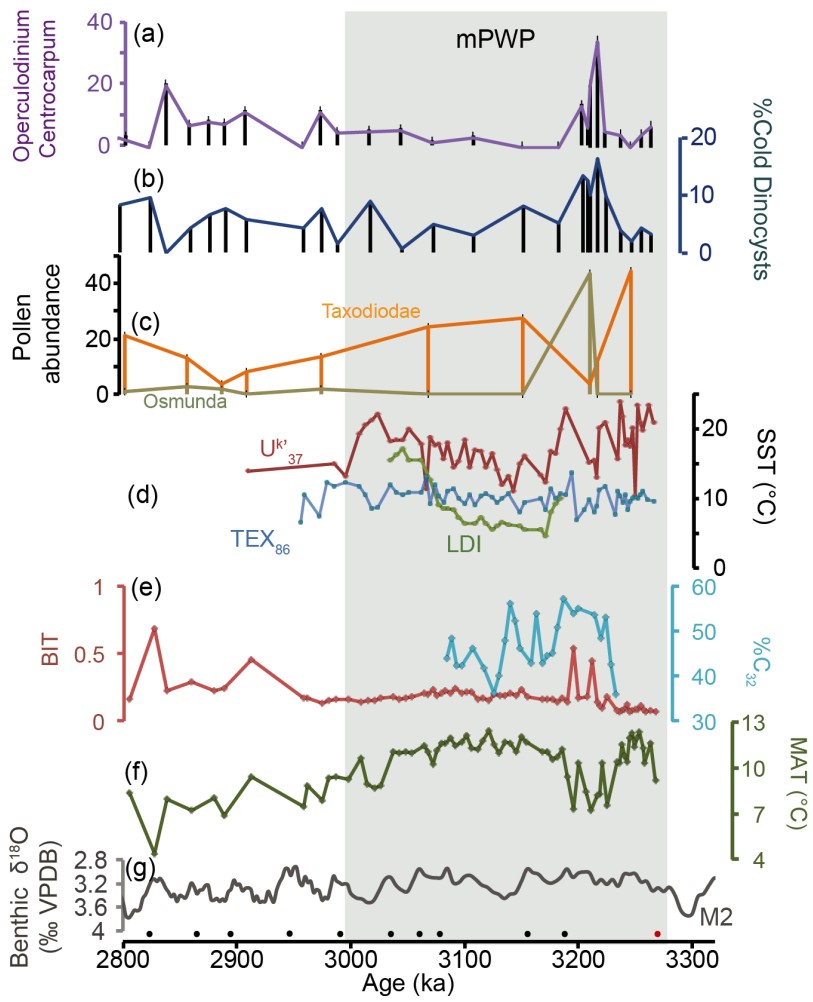

Fig. 6. Climate proxy records for the southern North Sea Basin around the mPWP. Age tie
points based on oxygen isotope stratigraphy (black) and biostratigraphy (red) are indicated.
(a) The relative abundance of *Operculodinium centrocarpum* expressed as a percent total
dinocysts as marker for the North Atlantic Current. (b) %Cold dinocysts, (c) pollen
abundances for Taxodioidae (orange) and Osmunda (dark yellow) as a percentage of the
pollen sum, (d) SST records based on the $TEX_{86}$, $U^{k'}_{37}$ and LDI proxies, (e) the relative input
of terrestrial organic material to the Hank sediments based on the Branched and Isoprenoid
Tetraether (BIT) index (from Dearing Crampton-Flood et al., 2018) and the input of fresh
water based on %C32 diol, (f) mean air temperature based on brGDGT-paleothermometry
(from Dearing Crampton-Flood et al., 2018), and (g) the benthic oxygen isotope stack of
Lisiecki and Raymo (2005).