# Peer review of "A new age model for the Pliocene of the Southern North Sea Basin: a multi proxy climate reconstruction"

_Climate of the Past, 2019_

## Short Comment (SC1) · Emily Dearing Crampton-Flood et al. · 10 May 2019

A comment to MIS M2 in the southern North Sea Basin:

A hiatus in sedimentation during MIS M2 was already suggested in papers that deal with the stratigraphy of the southern North Sea Basin. Studies by Head (1998), De Schepper et al. (2009, Geological Magazine) and Louwye et al. (2004, 2010, Geological Magazine) place the Belgian (Kattendijk, Lillo, Poederlee Fm) and English (Coralline Crag, Red Crag) Pliocene formations into one coherent stratigraphy. In De

Schepper et al. (2009) and Louwye et al. (2010), MIS M2 is identified as a sequence boundary, de facto a hiatus, in the southern North Sea Basin. These papers have not been taken into account, but would mostly support the conclusions here. See also the stratigraphic summary of De Schepper and Mangerud (2018, Norwegian Journal of Geology, Figure 7), which compares the northern North Sea Utsira Formation with Pliocene deposits in Iceland, England and Belgium. For L440: Rather than comparing with the Norwegian Sea record, it would be more relevant to compare here with records from the southern North Sea Basin (England, Belgium).

A comment to the mPWP in the southern North Sea Basin:

The Poederlee and Lillo Formation correspond to the interval 3.2–2.7 Ma. The paleoenvironmental information from those formations (De Schepper et al. 2009; Louwye et al. (2010) would be a valuable addition to the interpretations from the Hank core and be a major step forward towards a comprehensive summary of the climate and environmental evolution of the North Sea Basin during the mPWP and Late Pliocene.

A comment to the influence of the NAC in the North Sea:

The influence of the NAC in the Hank record is not convincing. Observing comparable SST variability is no proof for a causal relation (L40, L655 onwards). The common factor between the North Atlantic and the North Sea may be via the atmosphere (i.e. NAO). Note that while the SST variability in the eastern North Atlantic and Norwegian Sea correspond to the NAC (Naafs, Bachem, Lawrence), the cited SST variability in the Iceland Sea is related to the EGC (Clotten et al. 2018) (L655–659). Furthermore, most water from the North Atlantic flows into the North Sea Basin from the north. But in the manuscript, it is claimed that the NAC has a direct influence on the southern North Sea Basin through the shallow connection in the south (Channel/Dover) (L622-624). While an open connection after MIS M2 is possible, it remains speculative. Certainly because the presence of O. centrocarpum (sensu Wall and Dale 1966) in the Hank core is considered as evidence for the NAC influence in the North Sea. This does not

have to be the case, and most likely it is not - this is a cosmopolitan species. It is true that in the modern North Atlantic, O centrocarpum sensu Wall and Dale (1966) (aka. cysts of Protoceratium reticulatum) can be considered as good indicator for the NAC (e.g. Harland et al. 2016 in Helyon and refs therein). It has been used as an indicator for the NAC in the Pliocene eastern North Atlantic, in the region where the NAC flows (e.g. De Schepper et al. 2009 Paleoceanography, 2013 PLoS One, Hennissen et al. 2014 Paleoceanography). But today, when the Channel is open, it is not a common species in the North Sea (Marret et al. 2003 RPP, Zonneveld et al. 2013 RPP). Given that O centrocarpum (sensu Wall and Dale 1966) is foremost a cosmopolitan species, tolerant to wide range of SST, SSS, nutrients, etc., its occurrence in the North Sea may not be a simple function of North Atlantic water inflow.

Minor comments

L127, L621: It is not impossible, but it remains speculation whether a connection was established after MIS M2. The connection was likely only temporarily opened during the Pliocene when SL was high (e.g. see more recent papers by Van Vliet-Lanoë et al. 2002; Gibbard and Lewin 2016, Geologica Belgica). L317–318: Barssidinium is not the best example for a (sub)tropical taxon, as it occurs in Iceland in the Pleistocene (e.g. Verhoeven et al. 2011, Paleo-3). L625–630: O. centrocarpum (sensu Wall and Dale 1966) is foremost a cosmopolitan species recorded from different environments and tolerant to wide range of SST, SSS, nutrients, etc. Its occurrence in the North Sea shelf environment is thus not necessarily evidence for NAC influence. L629: Boessenkool et al. (2001) studied surface sediments offshore SE Greenland. The study does not provide evidence for a relationship between O. centrocarpum and the NAC. Please use more appropriate references.

---

## Referee Comment (RC1) · Anonymous Referee #1 · 12 Jun 2019

A well written paper which presents a multi-proxy based reconstruction of Pliocene marine sea surface temperatures and terrestrial climate of the Southern North Sea Basin. The core was taken in a marginal marine setting, and the interpretation particularly of the lipid biomarker proxy record is difficult as the signal is influenced by a multitude of marine and terrestrial factors. The authors are generally aware of the potential implications and the manuscript contains a thoughtful and careful discussion of the potential controls and limitations. However, the authors final conclusion about marine and terrestrial climate evolution during the mid Piacenzian warm period (mPWP) and

reorganization of the North Atlantic Current does not seem to be supported by data. I disagree with the statement that lipid biomarker and palynology-based temperature proxies suggest a stable warm climate during the mPWP. In fact, the lipid biomarker shows mean annual temperatures ranging from 7 to 12°C during the mPWP (Fig. 6) and only between 3.17 Ma and 3.1 Ma a plateau. No conclusion on terrestrial climate variability during the mPWP can be drawn from the terrestrial pollen and spores as this 300 ka long interval is only covered by 4 samples. TEX86, which has been interpreted as representing winter temperatures, shows relatively constant temperatures during the mPWP, while UK37 indicates very variable summer SSTs. However, the authors point out that the high values could in part be caused by freshwater algae that have little or no correlation with temperature. In fact, there are several indicators suggesting that at least some of the major changes in proxy values are controlled by the changing depositional environment and associated fluvial input. The statement of an overall "asynchronous shift of marine and terrestrial climate" indicating a reorganization of the NAC during the mPWP appears to be an unnecessary over-interpretation of the otherwise very interesting data.

Minor comments: Line 18 and 48: Be consistent with mPWP which is called in the abstract mid-Pliocene Warm Period and Introduction mid-Piacenzian Warm period. As the mPWP is part of the late Pliocene, the latter seems to be more appropriate.

Line 196: dinocysts and sporomorphs (or pollen and spores) were counted

Line 260 ff.: Can the lipid biomarker method part be shortened?

Line 327 and figures: "Heather" which represent the actual plants of the family Ericaceae is better than "heath" which normally means the entire heathland habitat.

Line 329: The authors counted approximately 200 pollen and spores per sample and excluded the bisaccate taxa from the pollen sum. I am wondering how many angiosperm pollen are actually left? Please provide more details on the total pollen sum (e.g. in Fig S1) and the pollen sum which has been used to calculate non-bisaccate

pollen percentages.

Line 513: Fig. 4d should be 3d

Line 425: Why do the authors refer to Donders et al, 2007? The palynological results from the Hank borehole seem to suggest the opposite in indicating the continuous presence of warm-temperate taxa (Fig. S1). Please discuss.

Line 605 and Fig. S1: The acme of Osmunda coincides with a decline of almost all other taxa in the pollen diagram. The authors state that Osmunda has been excluded from the pollen sum, and I am struggling to see which other taxa increased. They do not seem to sum up to 100% at ca. 306 m.

Fig.2 and Fig 3: Rephrase in figure captions "The intervals corresponding to A, B C depth discussed in the text are indicated" and provide keys to colours (e.g. marine/deltaic etc).

Fig 4. Provide proper depth scale instead of arrow.

Fig. 6 Add depth scale to age scale to allow better comparison across figures.

―――――――――――――――

---

## Referee Comment (RC2) · Anonymous Referee #2 · 11 Jul 2019

Interesting data are presented, however, the manuscript needs major revision. It is not clear what the data actually tell us, and why. One major issue is that it is not clear what the authors actually want to present. The tittle suggest that the main aim of the paper is to present a new age model and the introduction is mainly written with a focus on the need for a new age model. Secondly, the title hints at an asynchronous relation between marine and terrestrial climate. Even though both marine and terrestrial data is presented I do not see a focus on an asynchronous relation reflected in the paper. The authors need to figure out if the aim is to present an age model paper or a climate

paper, and structure the paper accordingly.

The discussion needs to be transferred from providing statements to become a discussion. What do your results show? How can they be explained/what can they tell? Why? What is the suggested mechanism? What is the argumentation for why that possible mechanism is most likely – and for others being less likely? When stating relations to other records, why don't you show some those records as well so that it can be evaluated?

The figure that presents the basis for the new age control is impossible to evaluate and needs to be redone. I would prefer to have the datasets correlated to each other shown underneath each other in the same direction so that the relations between the ups and downs of the two records are immediately visible. I would also like to see how the two records (LR04 and yours) match after correlation, showing both records towards age.

Some of the figures present a lot of data in a small format. Check font sizes and make sure they are readable.

Do you use the d13C data at all?

Comments that emerged while reading the manuscript:

L23: ….costal zones, linking the costal and continental climate evolution, are lacking.

L30: … stable oxygen isotopes…

L57: Fewer and less well constrained temperature records exist for the terrestrial realm (Zagwijn, 1992; Salzmann et al., 2013), however, they all indicate that climate was warmer than present. Why do you consider them to be less well constrained? Should be specified – because of age control? Because of other issues?

L106: I would be more careful than to say that the infaunal species are unaffected. They can trace large scale climate inflicted changes equally well as epifaunal species, e.g. detect brine signals during glacial times (e.g. Dokken et al., 2013). Furthermore,

you discuss a potential fresh water influence later on, so some inconsistence here.

L118: What is relative land cover?

L134: Your site is very close to the coast and I would expect much lower water depths than in the central basin. Do you have any idea about what the water depth was at your site? In such a shallow setting the water depth have implications for how to interpret your records.

L145: Why is it an advantage that you get a smoothed record?

L177: Did you rerun any of the samples you consider to be outliers? What is the argumentation for this choice for removing data point? It seems a bit arbitrary. What will the record look like if you include all measurements? What seems to be outliers can very well be true values if it just looks like the values don't belong.

L250: What about C37:4 as a fresh water indicator? Do you see similar changes there? I see that you state later that no C37:4 is present. Implications? Why do you see changes in one of your freshwater indicators but not in all?

L255: extracted and processed for what?

L283: What/why was it challenging?

L290: Here you state a variability of approximately 1‰ while based on the information above(L285 and onwards) its closer to 2.6‰ (or 0.9 to 1.8‰ if you selectively look at specific transitions). And why would you expect the d13c variability (in amplitude) to match the d18O variability?

Line 336: Lipid biomarkers and proxies. Why do you include "and proxies" here? The isotopes, palynology and biomarkers are all proxies.

L347: This is a very large range. Given your choice regarding "outliers" for the oxygen isotope record, why do you keep the biomarker results as measured? Supports the feeling I get that the removal of some isotope measurements are done a bit randomly

[Figure]

– even though you have a set way of defining which points you removed.

L355: selected

L374: Why do you get a stronger terrestrial influence towards the Pleistocene? Increased glacial erosion?

Line 411: MAT. Is this short for mean annual temperature? Not mentioned before, needs to be defined. Since you refer to your 1018 paper I assume the record is published and the method is described there, however, I am still curious about how certain you are regarding the absolute values presented given that your record includes extinct species, e.g. Sciatopitys? You refer to several other terrestrial profiles – can these be shown for comparison? If they lack age control, can you really link them to your record/state that it is the same?

L438: M2 is very pronounced in some records, but there are also several sites where its less pronounced, making it an enigmatic period with large uncertainties related to the magnitude of the "glacial" event. Risebrobakken et al., 2016 consider the possibility of a hiatus as an explanation for the lack of the most extreme signature but also that it might have been a less extreme event than expected from e.g. LR04.

Line 443: I cannot see that the North Sea is mentioned specifically in Miller et al., 2011? Overall, estimates of sea level change during M2 varies tremendously in literature. This should be acknowledged. Also take into account the findings of Raymo et al., 2018 where they conclude that for the Pliocene, geochemical sea level proxies currently carry uncertainties too large to allow any meaningful ice volume (hence sea level estimates).

LL461-472: You should not forget that your data is from a very shallow site and hence cannot be expected to reflect the same absolute values as the intermediate and deep water records from Risebrobakken and LR04. How do you physically transfer the suggested freshwater signal from the rivers to the bottom of the North Sea? A river signal

will be much less dense than a marine water mass and lay as a fresh lid on top of the denser water mass. This is one example of where knowledge of the paleodepth of your site is key to be able to make trustworthy interpretations of your data.

L479: The difference in amplitude of change between the global deep water stack and your shallow regional/costal site does not prove that the area is more sensitive to climate disturbances. Furthermore, what do you mean by climate disturbances? This is an empty term.

This section is also an example of statements without discussion. You should check this throughout and make sure that your discussion is a discussion and not just statements. Discuss your results, potential explanations, reasons for why one or the other potential explanation is more or less likely, and in the end conclude on what you find to be the most likely solution and why. The mechanisms are important. E.g. how do you physically make the fresh water reach the bottom of the North Sea in order to explain light benthic isotope values by fresh water influence.

L496: When you say that you tune the warmer periods, what exactly do you mean by that? I would never use the maxima or minima as tuning point between records, but rather go for transitions, since the character of the records you tune is bound to be different.

L505-513: From figure 5 it is impossible to evaluate the correlation between LR04 and your record and hence the basis for your age model. This figure needs to be improved. One suggestion would be to show the record so that you read them in the same way. Another thing I would require to see is a direct comparison the two isotope records vs age, following your new tuning of the record.

L527-528: I would expect that a dominance of deeper dwelling organisms influencing the GDGT data will provide colder temperatures than what you will expect for the surface, even if the water depth is shallow. At what depth is the thermocline located at the site today, annually and during summer? If it's a winter signal as you argue, what is

the difference between your values and the present winter temperatures (looks to be around 6$^{\circledR}$C according to WOA18)? Or even better, can you compare your temperature datasets to present day measurements from the same proxy in the same site/close by? Same for alkenones (L543) and LDI?

L540: Can you show a comparison towards some of these other records? All of these would relate to bottom water temperatures at your site – again, even if it is shallow there will be a clear difference between the top surface and the bottom water, especially if there is any seasonal biases. What does the temperature profiles look like today?

L550: Okhotsk and Rhode Island are quite different settings from your site. Are there any more local relevant studies to take into account?

L624: What is the present ocean circulation regime of the area? Should be presented in the introduction part of the paper. How large a fraction of the NAC enters the North Sea through the English Channel relative to north of Great Britain? Given the Pliocene geography of the area how different do you expect the circulation regime to have been?

L639: Why? How? The full section here where you link the Zagwijn data to your data without showing them and without really making it clear if you can or cannot do this seems speculative to me.

L652/L656: Can the variability be linked to the colder and warmer isotope stages? Does your variability compare to the changes seen in these other records?

L661: The variability discussed by Bachem et al., 2017 and linked to gateway changes in not related to mPWP.

L663: The freshwater influence suggested for Clotten et al. 2018 cannot be used as direct support for fresh water influence at your site.

L684-687: What is the argumentation and reasoning behind this statement? It is not clear how you support this conclusion.

L703: If this is the case you should show it. All the records you refer to are available online and can be plotted together with your data to document this statement. I would also like to see this relative to LR04 and your own d18O record, with the individual isotope stages visualized.

---

## Author Response (AR1)

Manchester, January 6, 2019

Dear Professor Winguth,

We would like to thank you, Stijn de Schepper, and two anonymous reviewers for their feedback on our manuscript entitled "A new age model for the Pliocene of the Southern North Sea Basin: evidence for asynchronous shifts of marine and terrestrial climate" [cp-2019-39], and the opportunity to revise our manuscript.

The main comments from reviewer #2 related to the age tuning and placing our Hank record in a larger spatial context. Both reviewers #1 and #2 brought up the concern that the title of the manuscript did not seem to reflect the progression of the discussion and conclusion sections. Finally, both reviewers and Stijn de Schepper raised issues with statements involving the influence of the North Atlantic Current (NAC) on the North Sea that we inferred using the dinocyst species *Operculodinium centrocarpum*. As can be seen in our revised manuscript and the author responses that we submitted previously, we followed most of the suggestions by the reviewers to further improve the manuscript. In addition, we have merged sections 4 and 5 of the Discussion to better reflect the focus of the paper, and also to limit repetition of the results. The integrated changes in the manuscript are visible by track changes.

The most important changes are:

- We have edited the title of the manuscript to "A new age model for the Pliocene of the Southern North Sea Basin: a multi proxy climate reconstruction", and restructured the discussion sections to better reflect the focus of the paper. The restructured discussion section combines the old Sections 4 and 5, which also leads to less repetition in the revised manuscript. (comments from reviewers #1 and #2)
- We have added paleoenvironmental information from the neighboring Poederlee and Lillo formations in Belgium (Section 4.2). This also aids us in providing an age constraint for the mid-Pliocene Warm Period. (comment from Stijn de Schepper)
- More speculative language is used regarding the influence of the NAC on the North Sea. We also adjusted our interpretation of the species *Operculodinium centrocarpum* as evidence for the influence of the NAC in our setting, as this species may not hold the same link with the NAC in the Pliocene North Sea as it does in the Pliocene North Atlantic. (Stijn de Schepper comment)
- Updated the figures, specifically Figures 4 (now Fig. 3) and 5, which now show a depth scale, following the remarks of reviewers #1 and #2. We have also added supplementary figures (S1 and S3) to the supplementary section, which show detailed tuning (Fig. S1), and comparison of the UK37 record with those of Naafs et al. (2010) and Lawrence et al. (2009; Fig. S3) to accommodate the comment of reviewer #2.

We hope that you find the revised manuscript acceptable for publication in Climate of the Past.

On behalf of all co-authors,

Yours sincerely,

**Emily Dearing Crampton-Flood**

**Reviewer #1 Response:**

A well written paper which presents a multi-proxy based reconstruction of Pliocene marine sea surface temperatures and terrestrial climate of the Southern North Sea Basin. The core was taken in a marginal marine setting, and the interpretation particularly of the lipid biomarker proxy record is difficult as the signal is influenced by a multitude of marine and terrestrial factors. The authors are generally aware of the potential implications and the manuscript contains a thoughtful and careful discussion of the potential controls and limitations.

A: We thank Anonymous Reviewer #1 for their kind comments about the manuscript and the proceeding comments which will help us in improving the manuscript. We reply to specific comments below.

However, the authors final conclusion about marine and terrestrial climate evolution during the mid Piacenzian warm period (mPWP) and reorganization of the North Atlantic Current does not seem to be supported by data. I disagree with the statement that lipid biomarker and palynology-based temperature proxies suggest a stable warm climate during the mPWP. In fact, the lipid biomarker shows mean annual temperatures ranging from 7 to 12°C during the mPWP (Fig. 6) and only between 3.17 Ma and 3.1 Ma a plateau. No conclusion on terrestrial climate variability during the mPWP can be drawn from the terrestrial pollen and spores as this 300 ka long interval is only covered by 4 samples.

A: Thank you for the comment. We agree with the reviewer that the proxy records indicate that the early part of the mPWP (>3200 ka) does not seem to have been very stable. However, this presumed instability may be an artefact of a disturbance in the record related to the recovery of the M2 or the M2 event itself (see discussion in response to Stijn de Schepper). Such a disturbance is also indicated by the coinciding peak in Osmunda pollen, which may indicate a sea level drawdown. In our discussion, we chose to focus on the period that we have been able to constrain with our new age model, and only covers a part of the mPWP, from ~3.18 Ma to ~3.0 Ma (black circles on Fig. 6), during which terrestrial climate appears stable. In the revised version of the manuscript, we will stress that the age tie-point represented by the red circle in Fig. 6 (~3.26 Ma) is based on biostratigraphy, and should therefore be interpreted with caution due to the large uncertainty associated with the last occurrence datum (LOD) of Melitasphaeridium choanophorum in the North Atlantic and the Nordic Seas, on which this point is based (see Dearing Crampton-Flood et al., 2018). Hence, we will also modify Fig. 6 to only shade the region that falls within the constraints of the updated age model based on  $\delta^{18}$ O presented in this study. The proxy records that we discuss in Section 5 of the paper are associated with this shaded region on which we have the best age control.

Furthermore, based on this comment and the remarks of Stijn de Schepper we will review the possible influence of the North Atlantic Current on the study site from the discussion in the revised version of the manuscript, adding more speculative language.

TEX86, which has been interpreted as representing winter temperatures, shows relatively constant temperatures during the mPWP, while UK37 indicates very variable summer SSTs. However, the authors point out that the high values could in part be caused by freshwater algae that have little or no correlation with temperature. In fact, there are several indicators suggesting that at least some of the major changes in proxy values are controlled by the changing depositional environment and associated fluvial input. The statement of an overall "asynchronous shift of marine and terrestrial climate" indicating a reorganization of the NAC during the mPWP appears to be an unnecessary over-interpretation of the otherwise very interesting data.

A: Thank you. In light of the comments made by yourself, Stijn de Schepper and the second anonymous reviewer we will amend the title of the revised manuscript to: "A new age model for the Pliocene of the Southern North Sea Basin: a multi proxy climate reconstruction". As mentioned above, we will tone down the discussion in the manuscript that you and the other reviewers have identified as being too

speculative, e.g. the influence of the NAC on the study area. In the revised manuscript, we will add the possible influence of the NAC as a hypothetical reason for the observed variability in SSTs recorded by lipid biomarkers.

Minor comments: Line 18 and 48: Be consistent with mPWP which is called in the abstract mid-Pliocene Warm Period and Introduction mid-Piacenzian Warm period. As the mPWP is part of the late Pliocene, the latter seems to be more appropriate.

A: We thank the reviewer for pointing out the inconsistency for the mPWP references. We will consistently refer to the mPWP as the mid-Piacenzian Warm Period in the revised manuscript.

Line 196: dinocysts and sporomorphs (or pollen and spores) were counted

A: Thank you, we will amend the sentence.

Line 260 ff.: Can the lipid biomarker method part be shortened?

A: We will shorten the specification and GDGT analysis method paragraph substantially, and will refer to Dearing Crampton-Flood et al. (2018; EPSL) for more details on the biomarker extraction methods and the GDGT analysis. The proceeding two paragraphs outlining the analysis of the long chain diols and alkenones have not yet been reported elsewhere and will be kept as is.

Line 327 and figures: "Heather" which represent the actual plants of the family Ericaceae is better than "heath" which normally means the entire heathland habitat.

A: Thank you for your suggestion, we will change the reference from 'Heath' to 'Heather' in the text and in Fig. 3G.

Line 329: The authors counted approximately 200 pollen and spores per sample and excluded the bisaccate taxa from the pollen sum. I am wondering how many angiosperm pollen are actually left? Please provide more details on the total pollen sum (e.g. in Fig S1) and the pollen sum which has been used to calculate non-bisaccate pollen percentages.

A: The pollen sum is not high, which is why we refrain from detailed paleoclimatological interpretations based on the pollen data. The main aim was to highlight the main quantitative trends, for which this pollen sum is adequate in a marine setting with no local vegetation variability. The pollen sum with and without bisaccates averages 250, and 60, respectively. The pollen sum includes all taxa except bisaccate conifers and Osmunda spores which were highly overrepresented in one sample. The caption of Figure S1 is incomplete, and we will adapt this. The two summary panels on the right side of the full diagram (Fig. S1) show both sums, the multi-colored panel excludes bisaccate conifers and is the primary percentage sum, and the grey shaded panel shows the bisaccate pollen as percent of the total terrestrial palynomorphs (the sum including all conifers and Osmunda). This way the abundances of all taxa can be compared without being affected by the potential transportation bias of bisaccate pollen. We will expand the caption of Figure S1 to clarify.

Line 513: Fig. 4d should be 3d

A: We assume there was a typo and the reviewer is referring to Line 413. Thank you, we will amend this.

Line 425: Why do the authors refer to Donders et al, 2007? The palynological results from the Hank borehole seem to suggest the opposite in indicating the continuous presence of warm-temperate taxa (Fig. S1). Please discuss.

A: In Donders et al. (2007), the majority of warm temperate taxa listed are shown to disappear at the Pliocene/Pleistocene transition. However, in these upland sites, the variable deltaic/fluviatile Pliocene deposits and incomplete preservation caused a hiatus of the uppermost Pliocene and lowermost Pleistocene deposits. The Hank site, however, is located in a shallow marine basin with a broader catchment and a regionally integrated signal, with a relatively more complete and reliable stratigraphy. However, the Hank site is not in contradiction to the land-based study where the earliest Pleistocene is probably not completely preserved and the last occurrences of warm-temperate taxa seem more abrupt (Donders et al., 2007). The clearest indication for this are the very low values of Taxodium-type pollen toward the top (Fig. S1). Tiglian deposits from the Netherlands, approximately dated to 2.0 Ma (Zagwijn, 1992; Quat. Sci. Rev.), still contain abundant Pterocarya, but only trace quantities of Taxodium-type and Carya (possibly reworked), and no Nyssa. From sequences in the central North Sea (Donders et al., 2018; Climate of the Past), it became clear that the earliest Pleistocene glacial-interglacials (MIS 102-92) do no longer contain Taxodium, Nyssa, or Carya, however this area receives sediment from southern Scandinavia (Eridanos), rather than the Rhine catchment. In summary, warm-temperate taxa disappeared in the earliest Pleistocene but not all at the same time, and most likely slightly above the level of the top of the Hank sequence, although warm-temperate taxa are clearly in decline as is seen in Fig. S1. We will rephrase the statement to clarify this point and add an additional reference.

Line 605 and Fig. S1: The acme of Osmunda coincides with a decline of almost all other taxa in the pollen diagram. The authors state that Osmunda has been excluded from the pollen sum, and I am struggling to see which other taxa increased. They do not seem to sum up to 100% at ca. 306 m.

A: This is explained in the point above on the pollen sum and caused by the incomplete description in the caption of Fig. S1. Osmunda increases and bisaccate taxa decline. At the same time (within the non-bisaccate total), Ericaceae, Alnus, and other fern spores increase relative to especially Taxodium-type.

Fig.2 and Fig 3: Rephrase in figure captions "The intervals corresponding to A, B C depth discussed in the text are indicated" and provide keys to colours (e.g. marine/deltaic etc).

A: We will amend the figure captions of Figs. 2 and 3 to read: 'Intervals 1, 2 and 3 discussed in the text are indicated by green (Early Pliocene), grey (mid-Pliocene), and purple (late Pliocene-early Pleistocene).'

Fig 4. Provide proper depth scale instead of arrow.

A: We will provide a depth scale in Fig. 4.

Fig. 6 Add depth scale to age scale to allow better comparison across figures.

A: We have attempted to add a depth scale to Fig. 6, however as the sedimentation rate is not continuous in this interval (Fig. 5e), we believe that adding a depth scale would create unnecessary confusion to the reader by making the figure too 'busy'. Instead, we will include the depth interval of the period covered by the tuning of oxygen isotopes to the LR04 stack (Fig. 5) to the figure caption.

**Reviewer #2 Response:**

Interesting data are presented, however, the manuscript needs major revision. It is not clear what the data actually tell us, and why. One major issue is that it is not clear what the authors actually want to present. The tittle suggest that the main aim of the paper is to present a new age model and the introduction is mainly written with a focus on the need for a new age model. Secondly, the title hints at an asynchronous relation between marine and terrestrial climate. Even though both marine and terrestrial data is presented I do not see a focus on an asynchronous relation reflected in the paper. The authors need to figure out if the aim is to present an age model paper or a climate paper, and structure the paper accordingly.

A: We thank the reviewer for their compliments on our dataset, and constructive comments on the manuscript. We appreciate their concern about what the data is telling us and why, and we add that this is quite a challenging dataset to interpret. The main focus of the paper, as you have determined from the first part of our title, is to present an age model for the Pliocene of the North Sea Basin. We have identified the need to construct an age model, which is missing from this region for the Pliocene, in order to place a variety of new and existing proxy records into context (e.g. L113-117). We agree that the discussion based on the asynchronous shifts of marine and terrestrial climate is not at the focus of the paper, and comes as part of a more speculative section near the end in section 5. Therefore, based on your comment and that of reviewer #1, we will amend the title of the paper to: "A new age model for the Pliocene of the Southern North Sea Basin: a multi proxy climate reconstruction", and adjust the focus of the discussion and conclusions accordingly.

The discussion needs to be transferred from providing statements to become a discussion. What do your results show? How can they be explained/what can they tell? Why? What is the suggested mechanism? What is the argumentation for why that possible mechanism is most likely – and for others being less likely? When stating relations to other records, why don't you show some those records as well so that it can be evaluated?

A: We are a bit confused by this comment, as we feel that we are doing this in sections 4 and 5. Section 4.1 discusses the logic and reasoning behind the age control that we present in Fig. 5, and involves the results from the palynology records, the seismic profile, the foraminifera isotopes, and the gamma ray logs ('What do the results show?', and 'How can they be explained?', and 'What is the mechanism?'). This discussion subsequently leads us to identifying the best interval suited for tuning to the LR04 benthic stack. Subsequently, section 4.2 presents a discussion on the potential confounding factors on the sea surface temperature (SST) records generated by the different lipid biomarkers, how the records should be interpreted, and why (i.e. explanation, comparison, mechanisms). We then use section 5 to discuss and interpret our multi-proxy climate records in the constrained age domain, and to place them in a global context (comparison and mechanisms). With regards to showing other records mentioned in the discussion section, see the specific reply to the comment below. Nevertheless, we will carefully read the discussion again and further clarify our lines of reasoning where possible in the revised manuscript.

The figure that presents the basis for the new age control is impossible to evaluate and needs to be redone. I would prefer to have the datasets correlated to each other shown underneath each other in the same direction so that the relations between the ups and downs of the two records are immediately visible. I would also like to see how the two records (LR04 and yours) match after correlation, showing both records towards age.

A: We feel like the gamma ray log and lithostratigraphic log on the figure are good additions for the reader to place the oxygen isotope record of Hank into context, since the foraminiferal isotopic

composition in such a shallow marine succession is dependent on other factors, such as depositional environment and hiatuses (see discussion in section 4.1). The lithostratigraphy and the gamma ray log can both give insight into these important processes (e.g. sharp inflections on the gamma ray log may indicate a hiatus). Similarly, the backdrop of the biostratigraphic age tying points is also useful to evaluate the oxygen isotope-based age model with the initial one presented in Dearing Crampton-Flood et al. (2018). Therefore, we feel that Fig. 5 warrants the addition of more information than just the age-tying points and the two oxygen isotope records. That being said, our revised evaluation of the periods where we suspect that ice volume changes may not play a big role (see discussion response to 'Do you use the d13C data at all?' below) will cause us to re-evaluate the tuning that we present in Fig. 5.

We will add a figure with the LR04 stack and the Hank oxygen isotope record on the same x-axis (age) to the supplementary information (Fig. S2) of our revised manuscript.

Some of the figures present a lot of data in a small format. Check font sizes and make sure they are readable.

A: We apologize if the labels in the figures are difficult to read. We will assess and optimize the font size of all figures where appropriate (Figs. 3 and 5 in particular).

**Do you use the d13C data at all?**

A: The  $\delta^{13}$ C and  $\delta^{18}$ O records give an indication about possible ice volume influences, given that the trends in  $\delta^{18}$ O and  $\delta^{13}$ C mirror each other (deep ocean concept). This is exemplified in the record of Noorbergen et al. (2015), who observed mirroring trends in the shallow marine Quaternary sequence from Noordwijk. When the trends between the  $\delta^{18}$ O and  $\delta^{13}$ C records do not mirror each other, the ice volume signal is less dominant due to other factors which overrule the signal such as freshwater influence, reworking, and diagenetic influences (the latter assumed to be minimal because only well-preserved foraminifera were picked). Therefore, including the  $\delta^{13}$ C record and comparing to the  $\delta^{18}$ O record will aid in demonstrating where ice volume signal is recorded in the Hank record and when it is not. We will include a sentence of where the two isotope records mirror each other in the results section (section 3.1) of the revised manuscript. Further, we will include a summary of this response into the discussion section of the paper in order to improve the evidence/statements/discussion that lead to the targeting of the tuned interval and therefore the creation of the final age model.

Comments that emerged while reading the manuscript:

L23: ....costal zones, linking the costal and continental climate evolution, are lacking.

A: Thank you for the suggestion, we will make the suggested change.

L30: ... stable oxygen isotopes...

A: We will make the suggested change.

L57: Fewer and less well constrained temperature records exist for the terrestrial realm (Zagwijn, 1992; Salzmann et al., 2013), however, they all indicate that climate was warmer than present. Why do you consider them to be less well constrained? Should be specified – because of age control? Because of other issues?

A: The temperature records presented by Zagwijn (1992) and compiled by Salzmann et al. (2013) are derived from pollen assemblages. Pollen based temperature estimates are generally less constrained

than that of e.g. SST reconstructions, as i) they present a temperature range rather than an absolute temperature estimate (Coope, 1970, Ann. Rev. Ent.; Mosbrugger and Utescher, 1997, Palaeogeo. Palaeoclima., Palaeoeco.), ii) they can be skewed toward growing season (Guiot, 1990, Palaeogeo. Palaeoclima., Palaeoeco.) and, iii) if based on terrestrial sediment sequences, can suffer from poor age control, as terrestrial outcrops are more difficult to date than marine sediment sequences. We will add this brief explanation to the appropriate section (introduction) in the revised manuscript.

L106: I would be more careful than to say that the infaunal species are unaffected. They can trace large scale climate inflicted changes equally well as epifaunal species, e.g. detect brine signals during glacial times (e.g. Dokken et al., 2013). Furthermore, you discuss a potential fresh water influence later on, so some inconsistence here.

A: We thank the reviewer for noticing this inconsistency. We will modify the sentence in the revised manuscript to read: "The depth habitat of endobenthic foraminifera in the sediment provides a moderate degree of shelter from disturbances such as reworking by bottom currents and freshwater input". This revised sentence provides a more accurate view of the role that the endobenthic foraminifera play in climate reconstructions.

**L118: What is relative land cover?**

A: We use the term relative land cover to describe the prevailing vegetation cover on the nearby continent during the time of deposition. We change this into 'prevailing vegetation' in the revised version of our manuscript.

L134: Your site is very close to the coast and I would expect much lower water depths than in the central basin. Do you have any idea about what the water depth was at your site? In such a shallow setting the water depth have implications for how to interpret your records.

A: According to the height of the clinoform as observed in the seismic profile (Fig. 4), the water depth in the interval ~300–200 m is approximately 80–100 m. The clinoform stacking suggests a continuous westward progradation of the coastal system, such that this location became progressively infilled by sediment. Note that this estimate does not refer to the downdrop which we associate with the (possible) M2 event or recovery thereof. We will add a few words to the end of the sentence to remind the reader that the Hank site in the Southern North Sea Basin is expected to have water depths lower than that predicted for the central basin. Later in the results and the discussion section where the seismic section is introduced and described we will specify the estimated water depth for the Upper Oosterhout formation (80–100 m).

**L145: Why is it an advantage that you get a smoothed record?**

A: The advantage of this method of core collection is that the 1 m resolution leads to smoothed records compared to the relatively expansive total record (404 m for the Hank Core). The possible disturbances generated from the dynamic environmental setting of the Hank site at this time will be smoothed, which makes it easier to reveal the more regional climate signal (albeit in lower resolution).

L177: Did you rerun any of the samples you consider to be outliers? What is the argumentation for this choice for removing data point? It seems a bit arbitrary. What will the record look like if you include all measurements? What seems to be outliers can very well be true values if it just looks like the values don't belong.

A: We realize that since the  $\delta^{18}$ O values of foraminifera recorded at the Hank site may not necessarily be attributed to ice volume signal alone, a different approach should be taken to identify outliers, given that freshwater input and reworking may affect the  $\delta^{18}$ O signal (see reply to earlier comment). Unfortunately, we did not rerun the samples we considered to be outliers. This hampers the interpretation of the record, especially in light of the extremely high and low  $\delta^{18}$ O values recorded (grey triangles in Fig. 2). Thus, we will remove the method of identification of outliers from section 2.2 in the manuscript, and update section 3.1 with revised values (ranges etc.) including the whole record. This will also involve replotting the  $\delta^{18}$ O data in Fig. 2, and Fig. 5 by extension.

The tuning for the age model is not expected to change largely because the majority of the tuned interval (296–206 m) does not contain that many outliers compared to the rest of the record (grey vs. black data points in Fig. 2). The tie points that are used particularly in the depth interval ~270–240 m will not change, particularly as this is the depth interval where a mirroring trend is seen between the  $\delta^{18}$ O and the  $\delta^{13}$ C records (see earlier response).

L250: What about C37:4 as a fresh water indicator? Do you see similar changes there? I see that you state later that no C37:4 is present. Implications? Why do you see changes in one of your freshwater indicators but not in all?

A: As we mentioned in L558-561 we did not detect the presence of the  $C_{37:4}$  alkenone in our samples. The absence of  $C_{37.4}$  in the mid-Pliocene (300–200 m) part of the interval does seem to fit with the other environmental indicators such as (i) low terrestrial/marine (T/M ratios), (ii) low BIT index values, and (iii) the presence of marine biomarkers such as Crenarchaeol and long chain diols (Figs. 2, 3). However, it is perhaps surprising that no  $C_{37.4}$  was detected in the interval '3' (Fig. 2), which corresponds to the late Pliocene/early Pleistocene, as the palynological data indicate a high T/M ratio and the presence of freshwater and brackish water algae species (L310-313). However, transect studies rom surface sediments in the Baltic Sea indicate that the percentage abundance of the tetraunsaturated C37:4 alkenone only appears higher (>15 %) at salinities around 8 psu and below (Bornholm Basin; Schulz et al., 2000; GCA; Kaiser et al., Org. Geochem.). Therefore, we interpret that the salinity change that affected the Hank Site as a result of a gradual change to a more estuarine environment did not change the salinity significantly enough to lower it below 8 psu. On the other hand, a very speculative reason behind this could be the affinity for the alkenone-producing organisms to the type of estuarine environment that the Hank site progressively became over the Plio-Pleistocene transition. However, until more is known about the specific organisms that produce these lipids, not much more can be said at this time.

L255: extracted and processed for what?

*A:* We will change this sentence into: 'lipid biomarkers were previously extracted from the sediments (*n*=155) and separated into polarity fractions according to …'

**L283: What/why was it challenging?**

A: The low abundance of foraminifera in the crag material made it challenging to pick in that interval. We will add this explanation to the revised manuscript.

L290: Here you state a variability of approximately 1‰ while based on the information above (L285 and onwards) its closer to 2.6‰ (or 0.9 to 1.8‰ if you selectively look at specific transitions). And why would you expect the d13c variability (in amplitude) to match the d18O variability?

A: We apologize for the confusion and will remove the reference to  $\delta^{18}$ O variability in L292-293, as the estimate in L286-287 is the more accurate representation of  $\delta^{18}$ O variability in this record. However, in

light of including all the  $\delta^{18}$ O values measured (as discussed above), the reported variability in the manuscript will change. Secondly, as far as we are concerned, we did not phrase the sentence to suggest that we expect the amplitude of  $\delta^{13}$ C variability to match the  $\delta^{18}$ O variability. We will rephrase this sentence as: 'Discounting the sample at 206 m, the variability in the  $\delta^{13}C_{cass.}$  record is approximately ~1 ‰ (Fig. 2d).'

Line 336: Lipid biomarkers and proxies. Why do you include "and proxies" here? The isotopes, palynology and biomarkers are all proxies.

A: We apologize for the confusion, the 'proxies' we refer to here refer to the specific proxies that are based on lipid biomarkers. We understand your confusion and will rename the section to read: 'Lipid biomarkers'.

L347: This is a very large range. Given your choice regarding "outliers" for the oxygen isotope record, why do you keep the biomarker results as measured? Supports the feeling I get that the removal of some isotope measurements are done a bit randomly– even though you have a set way of defining which points you removed.

A: Thank you for the comment. We agree that the Uk37 range we have calculated throughout the core is very large. We indeed mention the large range of reconstructed SSTs based on Uk37 in L347-353, L564-575, and L651-671, where we also compare the data with other SST records based on Uk37 in the North Atlantic, which also show a large range in SSTs. This is in contrast to the  $\delta^{18}$ O data, for which the range of variation is not reported elsewhere. We feel it is appropriate to plot the samples considered 'outliers' in the record so the reader can judge for themselves. This is why we have initially plotted those  $\delta^{18}$ O outliers in grey in Fig. 2c in our discussion paper, and also why we will include all  $\delta^{18}$ O measurements in our record in Fig. 2 and 5 in the revised manuscript.

**L355: selected**

A: We will not follow this suggestion as we believe this to be the correct usage of the term 'select' in this context.

L374: Why do you get a stronger terrestrial influence towards the Pleistocene? Increased glacial erosion?

A: The stronger terrestrial influence towards the Pleistocene at this site is most likely a result of sea level drawdown caused by the increasing volume of ice build-up in the northern hemisphere. The Hank site is a shallow marine succession which borders on the area between river/delta deposits and marine sands (Fig. 1 in manuscript), thus the progradation of the Rhine-Meuse River during this time would have brought more terrestrial material to the Hank site and contributed to the changing depositional environment during the transition into the cooler Pleistocene.

Line 411: MAT. Is this short for mean annual temperature? Not mentioned before, needs to be defined. Since you refer to your 1018 paper I assume the record is published and the method is described there, however, I am still curious about how certain you are regarding the absolute values presented given that your record includes extinct species, e.g. Sciatopitys? You refer to several other terrestrial profiles – can these be shown for comparison? If they lack age control, can you really link them to your record/state that it is the same?

A: MAT is indeed short for mean annual temperature, which we will specify in our revised manuscript. The MAT record has been published in Dearing Crampton-Flood (2018) and is based on temperaturesensitive membrane lipids of soil bacteria (branched GDGTs), which, in principle, yield absolute temperature estimates. A thorough discussion on the reliability of the absolute MAT values can be found in the same paper. Note that we have not reconstructed temperature using pollen distributions. We also note that species Sciatopitys is considered a relict taxon as it is not yet extinct (occurs in Japan). For a discussion regarding the age correlation between terrestrial profiles, see the discussion in the author responses to reviewer #1.

L438: M2 is very pronounced in some records, but there are also several sites where its less pronounced, making it an enigmatic period with large uncertainties related to the magnitude of the "glacial" event. Risebrobakken et al., 2016 consider the possibility of a hiatus as an explanation for the lack of the most extreme signature but also that it might have been a less extreme event than expected from e.g. LR04.

A: We have indeed considered whether the M2 event is a less extreme event than in some of the other areas of the world. However, there are many papers dealing with the stratigraphy of the North Sea that suggest a hiatus during MIS M2, particularly in the southern North Sea Basin (De Schepper et al., 2009, Geol. Mag.; Louwye et al., 2010, Geol. Mag.). See also the comment that Stijn de Schepper posted in the discussion. We also consider the occurrence of a hiatus during M2 at the Coralline Crag formation which is close to the Hank site (Fig. 1, manuscript), to explain the sequence boundary in the seismic profile at Hank. The position of this boundary indicates that the hiatus covers most of the M2 at the Hank site. We will expand the discussion in the revised manuscript to include the evidence for a hiatus over the M2 event in the adjacent sites in England and Belgium to support our interpretation. We will also add that Risebrobakken et al. (2016) considered that the M2 may have been a less extreme event in this region compared to other regions in the world where it is more pronounced, to provide the reader with a more nuanced view on the evidence for these different hypotheses on the M2 event in this region.

Line 443: I cannot see that the North Sea is mentioned specifically in Miller et al., 2011? Overall, estimates of sea level change during M2 varies tremendously in literature. This should be acknowledged. Also take into account the findings of Raymo et al., 2018 where they conclude that for the Pliocene, geochemical sea level proxies currently carry uncertainties too large to allow any meaningful ice volume (hence sea level estimates).

A: We apologize for the confusion; we will remove the specific reference to the North Sea in this sentence. We agree that the estimates for sea level change in the literature that correspond to the M2 event vary greatly. We also thank the reviewer for directing our attention to Raymo et al. (2018), and we will modify the sentence to read: 'There is evidence for a large global sea level drawdown (estimates of 70 m; Miller et al., 2011) during the M2, however large uncertainties in the estimation of ice volume prohibit any meaningful estimates of sea level for the Pliocene using the stable isotope measurements of foraminifera (Raymo et al., 2018).' However, the estimation of water depth at the Hank site is based on the seismic profile, and is approximately 80–100 m (see reply to earlier comment). Other estimates indicate that Pliocene sea level in the North Sea Basin was approximately 60–100 m (Overeem et al., 2001; Basin Res.), which agrees with our interpretation. These estimates however do not provide any insight into the sea level drawdown before that, associated with the possible M2 event or recovery of (corresponding to the sequence boundary in Fig. 4). We will include all relevant discussion on this point and references into the revised manuscript.

LL461-472: You should not forget that your data is from a very shallow site and hence cannot be expected to reflect the same absolute values as the intermediate and deep water records from Risebrobakken and LR04. How do you physically transfer the suggested freshwater signal from the rivers to the bottom of the North Sea? A river signal will be much less dense than a marine water mass and lay as a fresh lid on top of the denser water mass. This is one example of where knowledge of the paleo depth of your site is key to be able to make trustworthy interpretations of your data.

A: We appreciate the comment and we agree that the absolute  $\delta^{18}$ O values cannot be expected to be comparable to those of other records from cores recovered from deeper depths in the open ocean. We also consider the study of Noorbergen et al. (2015), who constructed a Quaternary age model from a similarly shallow succession in the North Sea, and found that the absolute  $\delta^{18}$ O values of a composite  $\delta^{18}$ O record of Bulimina aculeata, Cassidulina laevigata, Cibicides lobatulus, and Elphidiella hannai, were comparable to the LR04 stack. Hence, we tried to adopt the same approach to anticipate the influence of freshwater input by only picking endobenthic foraminifera in the Hank sediments (cf. Noorbergen et al., 2015). Hank is located relatively closer to the mouth of the paleo Rhine, and thus is likely associated with a shallower water depth (see response to previous comment for paleodepth interpretation of 80–100 m), which we believe is reflected in the large range of  $\delta^{18}$ O values, and thus likely also the offset compared to the LR04 stack. We will add the paleodepth interpretation to the revised manuscript. With regards to the transport of fresher water to lower depths in the North Sea, we note that the North Sea is a high energy system where wave action and winnowing contribute to the mixing of freshwater input in the relatively shallow water column (Charnock et al., Understanding the North Sea System, Springer Science and Business Media). We will include this explanation in the revised manuscript.

L479: The difference in amplitude of change between the global deep water stack and your shallow regional/costal site does not prove that the area is more sensitive to climate disturbances. Furthermore, what do you mean by climate disturbances? This is an empty term. This section is also an example of statements without discussion. You should check this throughout and make sure that your discussion is a discussion and not just statements. Discuss your results, potential explanations, reasons for why one or the other potential explanation is more or less likely, and in the end conclude on what you find to be the most likely solution and why. The mechanisms are important. E.g. how do you physically make the fresh water reach the bottom of the North Sea in order to explain light benthic isotope values by fresh water influence.

A: The larger amplitudes ( $\Delta \delta^{18}$ O > 2.5 ‰) observed in the Hank record are not the result of a dominant single climate factor such as ice volume. This is in contrast to case in deep ocean settings. At the Hank site, freshwater influence is more likely to lead to the large observed amplitude in  $\delta^{18}$ O. We will therefore adjust all language referring to 'climate disturbances' in the revised manuscript, and include the interpretation of the large amplitude as a result of periods where the Hank site was particularly sensitive to freshwater input, probably deriving from the proto Rhine-Meuse system. A literature search shows that similar observed amplitudes and associated low benthic and planktic  $\delta^{18}$ O values were recorded in marine sediments from the Ionian Sea, and were interpreted as freshwater influence that coincided with sapropel deposition (Schmiedl et al., 1998; Paleoceanography). We will inspect the discussion section carefully and include a more thorough discussion into this perceived freshwater input, with appropriate evidence and references. Further, the interpreted paleodepth (80–100 m) during the mPWP period (~300–200 m) will be added to the revised manuscript which leads to a better supported discussion on why the Hank site was sensitive to fresh water input, due to its shallow nature.

L496: When you say that you tune the warmer periods, what exactly do you mean by that? I would never use the maxima or minima as tuning point between records, but rather go for transitions, since the character of the records you tune is bound to be different.

A: We realise that we have not followed the method that is mostly used for age model constructions, however, the coastal location of the Hank site requires a slightly different approach, as is outlined in the manuscript (section 4.1). The combination of a relatively weak influence of the Atlantic and the potential occurrence of hiatuses during cooler phases, which could also include (part of the) transitions, has led us to use the warmer periods for tuning. Warmer periods in the record are identified as the maxima in the  $\delta^{18}$ O record (Pliocene interglacials: G17, G15, K1 etc. on LR04 stack). Furthermore, see the discussion above on the sampling resolution (1 m) of the Hank borehole, which may preclude the capture of any true inflection point or transition in the  $\delta^{18}$ O record.

As discussed above, we will identify the intervals of the record with opposing or mirroring trends in  $\delta^{18}$ O and  $\delta^{13}$ C and only use those for tuning the warm intervals to. This will lead to changes the discussion section and Fig. 5. We will therefore include this discussion, and a revised and updated form of Figure 5 in the manuscript.

L505-513: From figure 5 it is impossible to evaluate the correlation between LR04 and your record and hence the basis for your age model. This figure needs to be improved. One suggestion would be to show the record so that you read them in the same way. Another thing I would require to see is a direct comparison the two isotope records vs age, following your new tuning of the record.

A: As promised in our reply to an earlier comment, we will include a figure in the supplement that shows the correlation between the LR04 and the  $\delta^{18}$ O record from Hank, on (A) separate depth/age scales and on (B) the same age scale (Fig. S2).

L527-528: I would expect that a dominance of deeper dwelling organisms influencing the GDGT data will provide colder temperatures than what you will expect for the surface, even if the water depth is shallow. At what depth is the thermocline located at the site today, annually and during summer? If it's a winter signal as you argue, what is the difference between your values and the present winter temperatures (looks to be around6 C according to WOA18)? Or even better, can you compare your temperature datasets to present day measurements from the same proxy in the same site/closeby? Same for alkenones (L543) and LDI?

A: The modern thermocline depth near the Hank site is ~30 m (Richardson and Pedersen, 1998; ICES J. Mar. Sci.). In winter the water column is more isothermal and in summer is more stratified (Richardson and Pedersen, 1998), so it is difficult to pinpoint what the integrated signal throughout the year would be. Nevertheless, water column and sediment trap studies indicate that the TEX86 signal in the sediments reflects that of the subsurface, i.e. 50–300 m, consistent with their role as ammonia oxidizing archaea (Church et al., 2010, Environ. Microbiol.; Schouten et al., 2012, GCA). GDGT-producing Thaumarchaeota also occur in deeper water layers (>300 m), however, they occur in lower abundances, and a clear mechanism that can explain their transport to the sediment is lacking (Wuchter et al., 2005; Paleoceanography). Given the shallow water column of the Pliocene North Sea, the influence of deep(er) dwelling organisms can be ignored. Regardless, a potential contribution of deeper dwelling organisms can be recognized by a high (>5) ratio of GDGT-2/GDGT-3 (Tyler et al., 2013; Glob. Planet. Change), which we calculate for our data and discuss on L529-535 of the discussion manuscript.

Present day TEX86 SST reconstructions for the North Sea range between  $4.1 - 9.1^{\circ}$ C using the TEX86H calibration of Kim et al. (2010; GCA). These reconstructed SSTs correspond to observed SSTs of  $10.3 - 11.3^{\circ}$ C (WOA18). Therefore, core stop sediments in the North Sea are likely to underestimate the true SST; however, this statement may only be partly accurate insofar as the error on the proxy is ~2.5 °C (Kim et al., 2010). The Pliocene data from Hank hovers around ~9–12 °C for most of the record, which is in agreement or slightly higher than present day reconstructed SSTs using the same calibration. If it is interpreted as a winter signal as we argue, then the Hank Pliocene SSTs are approximately 3–6 °C higher than modern (van Aken, 2008; J. Sea Res.), which is in agreement with the data present in the manuscript in section 4.2.

Present day UK37 SST reconstructions for the North Sea: No suitable sites were found in the compilation of Tierney and Tingley (2018) of Uk37 recorded in surface sediments. The North Sea is a relatively underrepresented area in the UK37 calibration. However, reconstructed SSTs using alkenones from surface sediments from the Skagerrak region near the opening to the Baltic Sea record SSTs of 10–12 °C, approximately 1–2 °C higher than observed annual SST, and resemble those of May-June SST more (Blanz et al., 2005; GCA). Thus there is evidence for Uk37 recording summer temperatures (coinciding with bloom periods of haptophytes) in the circum-area North Sea.

Unfortunately, the core top calibration for the LDI SST proxy does not include core tops from the North Sea region. Therefore, we cannot compare our record to present day estimates of North Sea SSTs using the LDI proxy. This is due to the fact that the LDI is a relatively new proxy (Rampen et al., 2012; GCA), so it is not yet widely applied, especially not in combination with TEX86 and the Uk37. Indeed, the dataset presented in this manuscript is among the first paleo multi-proxy application studies of this proxy.

We will include a version of this discussion and comparisons of proxy-derived SSTs to modern SSTs in the revised version of the manuscript.

L540: Can you show a comparison towards some of these other records? All of these would relate to bottom water temperatures at your site–again, even if it is shallow there will be a clear difference between the top surface and the bottom water, especially if there is any seasonal biases. What does the temperature profiles look like today?

A: The temperature estimates of Wood et al (1993), Kuhlmann et al (2006), Johnson et al (2009) and Williams et al (2009) are not records of temperature over the whole of the Pliocene or mPWP intervals, but rather represent 'snapshots' of temperatures within these intervals. Thus it is not possible to plot our SST records and their reconstructed temperatures to directly compare SST evolution. However, it is possible to annotate Figures 2 and/or 6 in order to show the reconstructed temperatures based on these references in order to compare absolute values of temperatures reconstructed using the various proxies for approximately the same time interval. We will annotate Figure 6 to include annotations to the reconstructed temperatures of Wood et al (1993), Kuhlmann et al (2006), Johnson et al (2009) and Williams et al (2009), or a select few of the temperature estimates.

L550: Okhotsk and Rhode Island are quite different settings from your site. Are there any more local relevant studies to take into account?

A: We assume that by 'relevant' the reviewer means closer to the study site. In the revised manuscript, we will discuss the study of Blanz et al. (2005; GCA), who determined Uk37 values in a transect from the North Sea to the Baltic Sea and determined that for the Baltic Sea proper, there was no relationship between the alkenone UK37 indices with SST, whereas only the samples from Skagerrak plotted within 1 C of the global calibration of Müller et al. (1998; GCA). We will therefore add this discussion to section 4.2 in addition to these two sites.

L624: What is the present ocean circulation regime of the area? Should be presented in the introduction part of the paper. How large a fraction of the NAC enters the North Sea through the English Channel relative to north of Great Britain? Given the Pliocene geography of the area how different do you expect the circulation regime to have been?

A: The ocean circulation regime of the North Sea Basin today is dominated by wind-driven processes. The prevailing westerly winds on the north-west European shelf lead to a cyclonic anti-clockwise circulation, with the main input of water into the North Sea being from the north (Sündermann 2003; Oceanologia). We will add this information into the introduction of the paper. In the modern system, the modelled transport estimates from the HYCOM model indicate that the mean English Channel inflow is only 0.16 Sv (1 Sv =  $10^6 \text{ m}^3$ /s) versus the total mean inflow at the northern boundary of 2.22 Sv (Winther and Johannessen, 2006; J. Geophys. Res.: Oceans). In addition, the English Channel inflow varies intraannually, being weaker in winter and stronger in summer (Winther and Johannessen, 2006), meaning that the English Channel does not represent an important input of Atlantic water into the North Sea in the modern day. Given that the English Channel may or may not have been totally established in the Pliocene (Funnel, 1996; Quat. Sci. Rev.), and more than certainly closed during the M2 event, the NAC inflow would have originated entirely from the northern boundary

during the Pliocene. See also discussion in the author response to the comment posted by Stijn de Schepper. Scant evidence for a Pliocene connection of the North Sea to the North Atlantic via the English Channel is observed in the Pliocene-age Coralline Crag fauna and flora, which indicate planktonic elements, as well as bryozoan-dominated deposits that bear resemblance to modern deposits (Funnel, 1996). We will include a brief discussion on this observed evidence for and against a Pliocene connection from the North Sea to North Atlantic via English Channel in the revised manuscript.

L639: Why? How? The full section here where you link the Zagwijn data to your data without showing them and without really making it clear if you can or cannot do this seems speculative to me.

A: We tentatively correlate the Zagwijn Taxodium-type curve to our MAT record in the text, however since the Pliocene stages defined by Zagwijn are probably incomplete (as we state), we cannot be 100% definite in our correlation. A more solid conclusion that we do make in the revised manuscript is the match between the stratigraphy concept of Zagwijn and the Hank data. We will include a brief description and discussion on this in the revised manuscript.

L652/L656: Can the variability be linked to the colder and warmer isotope stages? Does your variability compare to the changes seen in these other records?

A: We will explore the possibility that the variability in the SST biomarker records may be related to colder and warmer isotope stages in the global stack and in the local oxygen isotope record at Hank.

L661: The variability discussed by Bachem et al., 2017 and linked to gateway changes in not related to mPWP.

A: Thank you for clarification. We will remove the reference to Bachem et al. (2017) in this instance.

L663: The freshwater influence suggested for Clotten et al. 2018 cannot be used as direct support for fresh water influence at your site.

A: We understand the reviewer's point, however L661-663 outline a possible reason that Clotten et al. (2018) determined as a contributor for the high variability in their Uk37 SST record in Iceland. We did not mean to translate their explanation as evidence for sea ice-derived fresh water influence at our site in the North Sea. To clarify, we will modify the sentence to read: 'The high variability of  $U^{K'}_{37}$  SSTs at the Hank Site during the Pliocene is most likely due to a combination of factors, including the shallow depth of the SNSB, potential changes in the direction and strength of the NAC, and varying freshwater influence.'

L684-687: What is the argumentation and reasoning behind this statement? It is not clear how you support this conclusion.

A: See comments and author response to Stijn de Schepper, and our response to earlier comments on the possible influence of the NAC. We have decided to re-evaluate the statements we make regarding the influence of the NAC in the North Sea Basin as a result of these comments. We will revise this sentence in the revised manuscript and make the potential influence of the NAC on our site more tentative. It will read: 'Regardless, the high variability and warming trend in two out of the three organic SST proxies in the Pliocene North Sea indicate that the area was very sensitive to environmental changes, of which the specific climatic drivers remain unclear.' L703: If this is the case you should show it. All the records you refer to are available online and can be plotted together with your data to document this statement. I would also like to see this relative to LR04 and your own d18O record, with the individual isotope stages visualized.

A: We will endeavour to obtain the SST data in order to plot a comparison of the variability of the records of Naafs et al. (2010), Bachem et al. (2017), and Lawrence et al., (2009), with our SST Uk37 record from the Hank site in order to show the similar variability in SSTs over the Pliocene. We will follow the suggestion to make a figure with the SST Uk37 record from Hank relative to the LR04 and the d180 record from Hank with the individual isotope stages visualized.

**Stijn de Schepper Response:**

A comment to MIS M2 in the southern North Sea Basin: A hiatus in sedimentation during MIS M2 was already suggested in papers that deal with the stratigraphy of the southern North Sea Basin. Studies by Head (1998), De Schepper et al. (2009, Geological Magazine) and Louwye et al. (2004, 2010, Geological Magazine) place the Belgian (Kattendijk, Lillo, Poederlee Fm) and English (Coralline Crag, Red Crag) Pliocene formations into one coherent stratigraphy. In De Schepper et al. (2009) and Louwye et al. (2010), MIS M2 is identified as a sequence boundary, de facto a hiatus, in the southern North Sea Basin. These papers have not been taken into account, but would mostly support the conclusions here. See also the stratigraphic summary of De Schepper and Mangerud (2018, Norwegian Journal of Geology, Figure 7), which compares the northern North Sea Utsira Formation with Pliocene deposits in Iceland, England and Belgium.

A: We thank Stijn de Schepper for the suggestions of the literature to be taken into consideration when discussing the MIS M2 in the revised manuscript. We will incorporate references to the relevant publications in the revised manuscript, particularly with reference to the base of the Poederlee Formation in Belgium which correlates with a sequence boundary Pia1 at approximately 3.21 Ma (De Schepper et al., 2009; Louwye et al., 2010; Geological Mag.). We agree that this would strengthen the interpretation of a hiatus taking place at the Hank site over the dramatic interval of MIS M2. We will add the appropriate references to the reference list.

For L440: Rather than comparing with the Norwegian Sea record, it would be more relevant to compare here with records from the southern North Sea Basin (England, Belgium).

A: Thank you for the comment. We agree that comparing the discussion in section 4.1.2 with records more adjacent to the Hank site in England and Belgium is a good idea, and will adjust our discussion in the revised manuscript accordingly.

A comment to the mPWP in the southern North Sea Basin: The Poederlee and Lillo Formation correspond to the interval 3.2–2.7 Ma. The paleoenvironmental information from those formations (De Schepper et al. 2009; Louwye et al. (2010) would be a valuable addition to the interpretations from the Hank core and be a major step forward towards a comprehensive summary of the climate and environmental evolution of the North Sea Basin during the mPWP and Late Pliocene.

A: We agree with Dr. de Schepper that a more detailed comparison of the environmental conditions during deposition of the Poederlee and Lillo formations in Belgium to the records in the Hank core would be a valuable addition to the paper. We will therefore incorporate and discuss the paleoenvironmental interpretations that are proposed in the above two manuscripts into the discussion of our revised manuscript and figures where appropriate.

A comment to the influence of the NAC in the North Sea: The influence of the NAC in the Hank record is not convincing. Observing comparable SST variability is no proof for a causal relation (L40, L655 onwards). The common factor between the North Atlantic and the North Sea may be via the atmosphere (i.e. NAO). Note that while the SST variability in the eastern North Atlantic and Norwegian Sea correspond to the NAC (Naafs, Bachem, Lawrence), the cited SST variability in the

Iceland Sea is related to the EGC (Clotten et al. 2018) (L655–659). Furthermore, most water from the North Atlantic flows into the North Sea Basin from the north. But in the manuscript, it is claimed that the NAC has a direct influence on the southern North Sea Basin through the shallow connection in the south (Channel/Dover) (L622-624). While an open connection after MIS M2 is possible, it remains speculative. Certainly because the presence of O. centrocarpum (sensu Wall and Dale 1966) in the Hank core is considered as evidence for the NAC influence in the North Sea. This does not have to be the case, and most likely it is not - this is a cosmopolitan species. It is true that in the modern North Atlantic, O centrocarpum sensu Wall and Dale (1966) (aka. cysts of Protoceratium reticulatum) can be considered as good indicator for the NAC (e.g. Harland et al. 2016 in Helyon and refs therein). It has been used as an indicator for the NAC in the Pliocene eastern North Atlantic, in the region where the NAC flows (e.g. De Schepper et al. 2009 Paleoceanography, 2013 PLoS One, Hennissen et al. 2014 Paleoceanography). But today, when the Channel is open, it is not a common species in the North Sea (Marret et al. 2003 RPP, Zonneveld et al. 2013 RPP). Given that O centrocarpum (sensu Wall and Dale 1966) is foremost a cosmopolitan species, tolerant to wide range of SST, SSS, nutrients, etc., its occurrence in the North Sea may not be a simple function of North Atlantic water inflow.

A: Thanks for the constructive comment, insight, and relevant literature concerning the presence of O centrocarpum (sensu Wall and Dale, 1966) and its modern and Pliocene occurrences. We refer to the author response to reviewer #2 who raised a similar concern, and also questions whether the influence of the NAC can be recorded in the North Sea using O. centrocarpum. We agree that there is scarce evidence for an open channel from the North Sea to the Atlantic via the English Channel, and that based on the evidence presented here by Stijn de Schepper, there may be little evidence to connect the trends in O. centrocarpum in the Hank record with influence of the NAC. Therefore, we will rewrite the discussion that links the occurrence of O. centrocarpum to the NAC in the revised version, making sure to refer to key literature and explain the reasons why using this strategy may be problematic in this North Sea setting.

Regarding the high SST variability observed in the Hank record, we believe that the similarities between the high variability recorded at the Hank site and other nearby records should be mentioned in the discussion, and the possible causes for high variability in the other records be outlined. However, as the link between the NAC and SST variability is too speculative and cannot be fully constrained, we will be more cautious in section 5.2 of the revised manuscript with regards to attributing the SST variability at the Hank site to any one specific factor, such as influence of the NAC.

Minor comments L127, L621: It is not impossible, but it remains speculation whether a connection was established after MIS M2. The connection was likely only temporarily opened during the Pliocene when SL was high (e.g. see more recent papers by Van Vliet-Lanoë et al. 2002; Gibbard and Lewin 2016, Geologica Belgica).

A: Thanks for this addition and references. We will definitely consider this in the revised version, and add more recent references when referring to the circulation and openings in the North Sea during the Pliocene.

L317–318: Barssidinium is not the best example for a (sub)tropical taxon, as it occurs in Iceland in the Pleistocene (e.g. Verhoeven et al. 2011, Paleo-3).

A: We will remove the reference to '(sub)tropical' in the sentence. We will also add the following sentence afterwards: '(Sub)tropical species like Lingulodinium machaerophorum, Operculodinium israelianum, Spiniferites mirabilis, Tectatodinium pellitum and Tuberculodinium vancampoae are missing at this depth.'.

L625–630: O. centrocarpum (sensu Wall and Dale 1966) is foremost a cosmopolitan species recorded from different environments and tolerant to wide range of SST, SSS, nutrients, etc. Its occurrence in the North Sea shelf environment is thus not necessarily evidence for NAC influence.

A: As per our reply to the major comment raised above, we will rewrite the discussion to tentatively connect the presence and trends in O. centrocarpum in the Hank record to the strength of the NAC in the revised manuscript. We do see an interesting feature in the O. centrocarpum record insofar as a large abundance (~40 %) that occurs at 305 m, directly after the high abundance of Osmunda spores and % Cold Dinocysts at 306 m. We interpret this increase in O centrocarpum as a possible restoration of (mostly) marine conditions at the Hank site after sea level drawdown which we interpret at 306 m. This period also correlated with low mean annual air temperatures (~ 6 °C), and decreased abundance of Taxodium-type pollen species (Fig. 6). We are not sure how best to interpret this, the best possible explanation we have at this stage is to connect the high abundance of O centrocarpum with the inflow of Atlantic Water when sea levels rose after the M2 event. We believe that the revised manuscript could benefit from some discussion on this observed feature in Section 5, we will make sure to refer to literature presented here about O centrocarpum in order to provide a more tentative and speculative explanation of this link.

L629: Boessenkool et al. (2001) studied surface sediments offshore SE Greenland. The study does not provide evidence for a relationship between O.centrocarpum and the NAC. Please use more appropriate references.

A: We will remove the references that tie the presence of O. centrocarpum to the NAC (see above) in the revised manuscript, thus the reference of Boessenkool et al. (2001) will also be deleted (and removed from the reference list).

**A new age model for the Pliocene of the Southern North Sea Basin: evidence for asynchronous shifts of marine and terrestrial climate a multi proxy climate reconstruction**

Emily Dearing Crampton-Flood1\*, Lars J. Noorbergen2, Damian Smits1, R. Christine Boschman1, Timme H. Donders4, Dirk K. Munsterman5, Johan ten Veen5, Francien Peterse1, Lucas Lourens1, Jaap S. Sinninghe Damsté1,3

1Department of Earth Sciences, Utrecht University, Utrecht, The Netherlands

[revised manuscript text omitted]
. does 145 not represent an important input of Atlantic water into the North Sea in the Pliocene. In addition to a main marine water supply via the North Atlantic, the Eridanos River, draining the Fennoscandian shield, and the proto-Rhine-Meuse River, draining North Western Europe delivered freshwater to the North Sea in the Pliocene (Fig. 1). The proto-Rhine-Meuse river system existed for a large part of the Pliocene, initially draining the Rhenish Massif and 150 finally later making a connection with the Alps in the latest Pliocene (Boenigk, 2002). During the Pliocene, the sediment supply by the Eridanos River system to the Ruhr Valley Rift system was limited, such that the Rhine-Meuse river system was the predominant supplier of sediments in the study area (Westerhoff, 2009). The water depth of the North Sea during the Pliocene and the Pleistocene was approximately 100-300 m in the central part of the basin (Donders et al., 55 2018), and approximately 60–100 m in the southern North Sea Basin (Overeem et al., 2001). Modern circulation in the North Sea is dominated by wind driven processes, which leads to a eyclonic anti-clockwise circulation. 
[revised manuscript text omitted]

Long chain diols used for calculation of the LDI index are below the detection limit in a large proportion of the Hank borehole. SSTs can be reconstructed for a select few samples in sections-intervals 1 and 3, and they show scattered temperatures in a range of 13 °C (Fig. 2e). The sediments in section-interval 2 contain enough diols to enable a semi-continuous SST reconstruction. The range of LDI SSTs in this section 2B is 4–18 °C (Fig. 2e). The record shows a strong warming trend of  $\sim 120-12$  °C from 295–263 m, coeval with the trend in the UK37 record (Fig. 2). The %C32 diol is generally high ( $\sim 36-57\%$ ) in 1 and 2 (Early-Mid Pliocene; Fig 34c), 
[revised manuscript text omitted]

| Formatted: Font: (Default) Times New Roman, 12 pt                                  |
|------------------------------------------------------------------------------------|
| Formatted: Font color: Auto                                                        |
| Formatted: Font: (Default) Times New Roman, 12 pt, Font color: Auto                |
| Formatted: Space After: 0 pt, Pattern: Clear                                       |
| Formatted: Font: (Default) Times New Roman, 12 pt, Font color: Auto                |
| Formatted: Font: (Default) Times New Roman, 12 pt, Font color: Auto                |
| Formatted: Font: (Default) Times New Roman, 12 pt, Not
Italic, Font color: Auto |
| Formatted: Font: (Default) Times New Roman, 12 pt, Font color: Auto                |
| Formatted: Font: (Default) Times New Roman, 12 pt                                  |
| Formatted: Font: (Default) Times New Roman, 12 pt                                  |
| Formatted: Font color: Auto                                                        |
| Formatted: Indent: Left: 0 cm, First line: 0 cm                                    |
|                                                                                    |

| Formatted: Space After: | 0 pt, Pattern: Clear |
|-------------------------|----------------------|
|-------------------------|----------------------|

| Formatted: Font color: Auto                                                        |
|------------------------------------------------------------------------------------|
| Formatted: Font color: Auto                                                        |
| Formatted: Font: (Default) Times New Roman, 12 pt                                  |
| Formatted: Font color: Auto                                                        |
| Formatted: Font: (Default) Times New Roman, 12 pt, Font color: Auto                |
| Formatted: Font: (Default) Times New Roman, 12 pt, Font color: Auto                |
| Formatted: Font: (Default) Times New Roman, 12 pt, Font color: Auto                |
| Formatted: Font: (Default) Times New Roman, 12 pt, Not
Italic, Font color: Auto |
| Formatted: Font: (Default) Times New Roman, 12 pt, Font color: Auto                |
| Formatted: Font: (Default) Times New Roman, 12 pt                                  |
| Formatted: Font: (Default) Times New Roman, 12 pt                                  |
| Formatted: Font: (Default) Times New Roman, 12 pt, Font color: Auto                |
| Formatted: Font color: Auto                                                        |

[revised manuscript text omitted]

| 1 | Formatted: Font: (Default) Times New Roman, Font color:
Auto |
|---|-----------------------------------------------------------------|
| Ч | Formatted: Font color: Auto                                     |

Pliocene climate: results from the Pliocene Model Intercomparison Projecta- Clim. Past, 9(1), 191, DOI:10.5194/cp-9-191-2013, 2013.

Head, M. J.: Modern dinoflagellate cysts and their biological affinities, in: Palynology: Principles and Application, Vol. 3, edited by: Jansonius, J. and McGregor, D. C., American Association of Stratigraphic Palynologists Foundation, College Station, Texas, USA, 1197–1248, 1996.

1190

- Head, M.J.: Pollen and dinoflagellates from the Red Crag at Walton-on-the-Naze, Essex:
   evidence for a mild climatic phase during the early Late Pliocene of eastern
   Englanda- Geol. Mag., 135(6), 803-817, 1998.
- Hennissen, J.A., Head, M.J., De Schepper, S. and Groeneveld, J.: Palynological evidence for a southward shift of the North Atlantic Current at ~2.6 Ma during the intensification of
   late Cenozoic Northern Hemisphere glaciation1, Paleoceanography and Paleoclimatology, 29(6), 564-580, https://doi.org/10.1002/2013PA002543, 2014.
- Hennissen, J.A., Head, M.J., De Schepper, S. and Groeneveld, J.: Dinoflagellate cyst paleoecology during the Pliocene–Pleistocene climatic transition in the North Atlantic. PalaeogeographyPalaeogeogr., Ppalaeocl\_imatology, palaeoecology, 470, 81-108, https://doi.org/10.1016/j.palaeo.2016.12.023, 2017.
  - Herfort, L., Schouten, S., Boon, J.P., Woltering, M., Baas, M., Weijers, J.W. and Sinninghe Damsté, J.S.: Characterization of transport and deposition of terrestrial organic matter in the southern North Sea using the BIT index2, Limnol. Oceanogr., 51(5), 2196-2205, https://doi.org/10.4319/lo.2006.51.5.2196, 2006.
- 1210 Heusser, L.–E. and Shackleton, N.–J.: Direct marinecontinental correlation: 150,000-year oxygen isotope-pollen record from the North Pacific, Science, 204, 837–839, https://doi.org/10.1126/science.204.4395.837, 1979.
  - Hodgson, G.E. and Funnell, B.M.: Foraminiferal biofacies of the early Pliocene Coralline Crag. Micropalaeontology of Carbonate Environments2, British Micropalaeontological Society Series. Ellis Horwood Ltd., Chichester, UK, 44-73, 1987.
  - Hopmans, E.C., Weijers, J.W., Schefuß, E., Herfort, L., Sinninghe Damsté, J.S. and Schouten, S.: A novel proxy for terrestrial organic matter in sediments based on branched and isoprenoid tetraether lipids.- Earth Planet. Sc. Lett., 224(1-2), 107-116, https://doi.org/10.1016/j.epsl.2004.05.012, 2004.

| 1220 | Hopmans, E | E.C.,  | Schouten,   | S.    | and    | Sinninghe | Damsté,     | J.S.:         | The | effect | of   | impr | oved |  |
|------|------------|--------|-------------|-------|--------|-----------|-------------|---------------|-----|--------|------|------|------|--|
|      | chron      | natog  | raphy on    | (     | GDG7   | Г-based ј | alaeoproxi  | es . O | rg. | Geoche | em., | 93,  | 1-6, |  |
|      | https:/    | //doi. | org/10.1016 | 5/j.c | orggeo | chem.2015 | .12.006, 20 | )16.          |     |        |      |      |      |  |

 Huguet, C., Schimmelmann, A., Thunell, R., Lourens, L.J., Sinninghe Damsté, J.S.: A study of

 1225
 the TEX86 paleothermometer in the water column and sediments of the Santa Barbara

 Basin, California, Paleoceanography, 22, https://doi.org/10.1029/2006PA001310,

 2007.

- IPCC, 2014: Climate Change 2014: Synthesis Report. Contribution of Working Groups I, II and III to the Fifth Assessment Report of the Intergovernmental Panel on Climate
   Change [Core Writing Team, R.K. Pachauri and L.A. Meyer (eds.)]a7 IPCC, Geneva, Switzerland, 151 pp.
  - Jacobs, Z.: Luminescence chronologies for coastal and marine sediments2- Boreas, 37(4), 508-535, https://doi-org.proxy.library.uu.nl/10.1111/j.1502-3885.2008.00054.x, 2008.

Jansen, H.S.M., Huizer, J., Dijkmans, J.W.A., Mesdag, C. and Van Hinte, J.E.;<del>, 2004.</del> The geometry and stratigraphic position of the Maassluis Formation (western Netherlands and southeastern North Sea)-, Netherlands Neth. Journal J. of GeosciencesGeosci., 83(2), pp.93-99, https://doiorg.proxy.library.uu.nl/10.1017/S0016774600020060, 2004.

Janssen, N., and Dammers, G.: Sample Processing for Pre-Quaternary Palynology1. Internal 1240 TNO Report, 2008.

[revised manuscript text omitted]

| Formatted: Font color: Auto                                                        |
|------------------------------------------------------------------------------------|
| Formatted: Font: (Default) Times New Roman, 12 pt                                  |
| Formatted: Font color: Auto                                                        |
| Formatted: Font: (Default) Times New Roman, 12 pt, Font color: Auto                |
| Formatted: Font: (Default) Times New Roman, 12 pt, Font color: Auto                |
| Formatted: Font: (Default) Times New Roman, 12 pt, Font color: Auto                |
| Formatted: Font: (Default) Times New Roman, 12 pt, Not
Italic, Font color: Auto |
| Formatted: Font: (Default) Times New Roman, 12 pt, Font color: Auto                |
| Formatted: Font: (Default) Times New Roman, 12 pt                                  |
| Formatted: Font: (Default) Times New Roman, 12 pt                                  |
| Formatted: Font: (Default) Times New Roman, 12 pt, Font color: Auto                |
| Formatted: Font color: Auto                                                        |
| Formatted: Font color: Auto                                                        |
| Formatted: Font color: Auto                                                        |
| Formatted: Font color: Auto                                                        |
| Formatted: Font color: Auto                                                        |
| Formatted: Font color: Auto                                                        |
| Formatted: Font: (Default) Times New Roman, 12 pt                                  |
| Formatted: Font color: Auto                                                        |
| Formatted: Font: (Default) Times New Roman, 12 pt, Font color: Auto                |
| Formatted: Font: (Default) Times New Roman, 12 pt, Not
Italic, Font color: Auto |
| Formatted: Font: (Default) Times New Roman, 12 pt, Not Italic, Font color: Auto    |
| Formatted: Font: (Default) Times New Roman, 12 pt, Font color: Auto                |
| Formatted                                                                          |
| Formatted                                                                          |
| Formatted: Font: (Default) Times New Roman, 12 pt                                  |
| Formatted: Font: (Default) Times New Roman, 12 pt                                  |
| Formatted                                                                          |
| Formatted: Font color: Auto                                                        |
| Formatted: Font color: Auto                                                        |
| Formatted                                                                          |
| Formatted: Font: (Default) Times New Roman, 12 pt                                  |
| Formatted: Font: (Default) Times New Roman, 12 pt                                  |
| Formatted                                                                          |
| Formatted: Font color: Auto                                                        |

[revised manuscript text omitted]